# MEDVR: ANNOTATION-FREE MEDICAL VISUAL REASONING VIA AGENTIC REINFORCEMENT LEARNING

**Zheng Jiang**[1,2*] **Heng Guo**[2,3*] **Chengyu Fang**[1,2*] **Changchen Xiao**[5] **Xinyang Hu**[5]
**Lifeng Sun**[1,4†] **Minfeng Xu**[2,3†]

[1]Tsinghua University [2]DAMO Academy, Alibaba Group [3]Hupan Lab, 310023, Hangzhou, China
[4]Key Laboratory of Pervasive Computing, Ministry of Education [5]Zhejiang University
jz24@mails.tsinghua.edu.cn, chengyufang.thu@gmail.com
sunlf@tsinghua.edu.cn, {gh205191, eric.xmf}@alibaba-inc.com

## ABSTRACT

Medical Vision-Language Models (VLMs) hold immense promise for complex clinical tasks, but their reasoning capabilities are often constrained by text-only paradigms that fail to ground inferences in visual evidence. This limitation not only curtails performance on tasks requiring fine-grained visual analysis but also introduces risks of visual hallucination in safety-critical applications. Thus, we introduce MedVR, a novel reinforcement learning framework that enables annotation-free visual reasoning for medical VLMs. Its core innovation lies in two synergistic mechanisms: Entropy-guided Visual Regrounding (EVR) uses model uncertainty to direct exploration, while Consensus-based Credit Assignment (CCA) distills pseudo-supervision from rollout agreement. Without any human annotations for intermediate steps, MedVR achieves state-of-the-art performance on diverse public medical VQA benchmarks, significantly outperforming existing models. By learning to reason directly with visual evidence, MedVR promotes the robustness and transparency essential for accelerating the clinical deployment of medical AI.

## 1 INTRODUCTION

Medical Vision-Language Models (VLMs) are increasingly applied to clinical tasks such as computer-aided diagnosis, report generation, and decision support (Chen et al., 2024b; Xu et al., 2025; Hartsock & Rasool, 2024). Achieving reliable performance in these tasks requires models to perform multi-step reasoning that combines fine-grained visual analysis with medical expertise. Recent advances have demonstrated that Reinforcement Learning with Verifiable Rewards (RLVR) can substantially bolster the reasoning capabilities of general-purpose VLMs (Guo et al., 2025). Inspired by this success, there is growing interest in adapting RLVR to endow medical VLMs with more robust reasoning abilities.

However, the prevailing RLVR paradigm remains almost exclusively confined to the textual domain, imposing fundamental limitations at odds with the demands of clinical practice (Lai et al., 2025; Pan et al., 2025; Dou et al., 2025). First, many diagnostic challenges, including the localization of subtle lesions, comparison of tissue densities, interpretation of hemodynamic flow, and quantification of anatomical structures, necessitate a level of fine-grained visual grounding that a text-only chain-of-thought cannot provide (Liu et al., 2025a). The absence of direct interaction with visual data during the reasoning process severely curtails the model's ability to exploit the rich information embedded in medical images. Second, this reliance on text-only reasoning renders models susceptible to visual hallucinations and language shortcuts, where answers are generated from textual priors while ignoring visual information (Huang et al., 2024). Such failures introduce unacceptable risks in safety-critical medical applications. In stark contrast, human clinicians dynamically interact with images, zooming, adjusting windowing, defining regions of interest, and measuring findings to formulate and validate hypotheses. This discrepancy motivates a central research question: ***Can we empower medical VLMs to perform complex reasoning by naturally interleaving textual deliberation with image manipulation tools, thereby substantiating key analytical steps with verifiable visual findings?***

---

*Equal Contribution, †Corresponding Author.

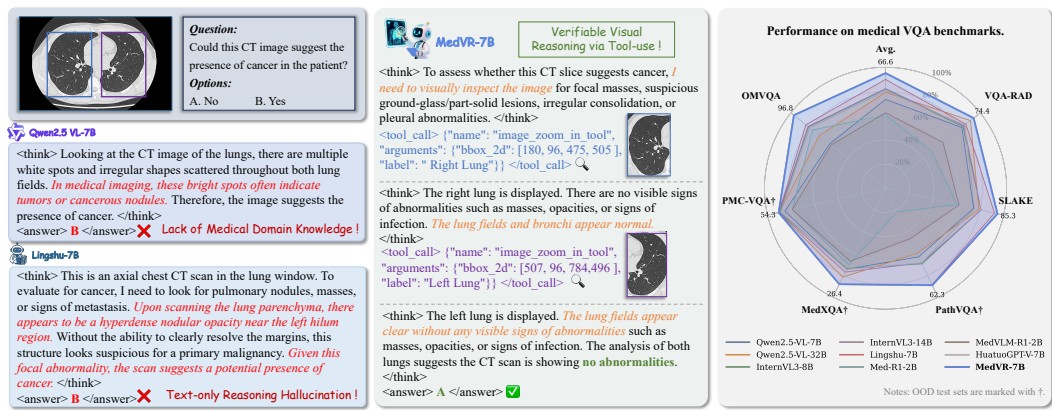

Figure 1: **MedVR enables verifiable visual reasoning via tool-use.** In contrast to the visual hallucinations prevalent in text-only reasoning (left), MedVR (middle) interleaves textual deliberation with active image manipulation to achieve precise visual grounding. This synergy results in superior accuracy and robustness across multiple public medical VQA benchmarks (right).

Although frameworks enabling interleaved language and tool-based image operations have emerged in general computer vision (Zheng et al., 2025; Su et al., 2025), their direct translation to the medical domain is fraught with challenges. The primary obstacle lies in the critical need for domain adaptation. General-purpose VLMs, trained on vast corpora of internet images, lack the specialized knowledge to connect medical imagery with clinical concepts and fail to generate reliable zero-shot localizations for subtle pathological findings (Xu et al., 2025; Kalpélbé et al., 2025). Consequently, they typically require extensive, fine-grained supervision of intermediate steps to learn meaningful visual grounding (Liu et al., 2025a). However, this requirement creates a paradox as the annotations needed for such intermediate supervision are notoriously scarce and expensive in medicine, rendering the training and validation of explicit visual reasoning process prohibitively difficult.

To bridge this gap, we introduce MedVR, an end-to-end reinforcement learning framework that, to our knowledge, is the first to achieve annotation-free visual reasoning for medical VLMs. As shown in Figure 1, MedVR empowers a VLM to seamlessly interleave textual chain-of-thought reasoning with the use of image-manipulation tool. It circumvents the need for supervisory labels by augmenting verifiable end-task rewards with two novel mechanisms that provide fine-grained, self-generated supervision. The first, Entropy-guided Visual Regrounding (EVR), leverages the model's intrinsic predictive uncertainty to guide exploration. By monitoring token-level entropy during inference, EVR identifies moments of high uncertainty, signaling ambiguity in grounding, and triggers targeted exploration from these states. This adaptive branching diversifies reasoning trajectories precisely where the model is least confident, encouraging it to "re-ground" its attention on the image before proceeding. The second mechanism, Consensus-based Credit Assignment (CCA), then synthesizes information from the ensemble of trajectories generated by EVR. By aggregating the grounding boxes produced across diverse rollouts, CCA distills a high-confidence consensus target. This consensus serves as a self-generated pseudo-label, providing a fine-grained supervisory signal for visual operations without any human annotation. In essence, EVR acts as a *prior* exploration strategy guided by uncertainty, while CCA provides a *posterior* supervision distilled from collective agreement. This synergy creates an entirely unsupervised curriculum for visual reasoning, where exploration is focused on critical decision points and in-batch consensus provides actionable feedback.

We conducted a rigorous and thorough evaluation to demonstrate the capabilities of MedVR. Our framework was benchmarked against a comprehensive suite of public medical Visual Question Answering (VQA) datasets. MedVR consistently achieves state-of-the-art results across both in-domain and out-of-domain scenarios, thereby demonstrating its generalizability and superior reasoning capabilities. Furthermore, a series of ablation studies validate the necessity and substantial impact of both EVR and CCA. Our principal contributions are summarized as follows:

(1) We introduce MedVR, the first end-to-end reinforcement learning framework that integrates visual and textual reasoning for medical VLMs, eliminating the need for costly intermediate supervision.

(2) We propose two novel, label-free mechanisms: Entropy-guided Visual Regrounding (EVR) for uncertainty-aware exploration and Consensus-based Credit Assignment (CCA) for self-generated supervision, which collectively create a self-supervising curriculum for fine-grained visual reasoning.

(3) We establish new state-of-the-art results on multiple public medical VQA benchmarks, showcasing MedVR's broad applicability, robustness, and enhanced reasoning capabilities in the medical domain.

## 2 RELATED WORK

**Medical Reasoning with Vision-Language Models.**   Vision-Language Models (VLMs) have shown remarkable potential in multimodal medical analysis by jointly reasoning over clinical texts and medical images (Hartsock & Rasool, 2024; Xu et al., 2025; Dai et al., 2025; Fang et al., 2026). Pioneering works, including LLaVA-Med (Li et al., 2023b), Med-Flamingo (Moor et al., 2023), and HuatuoGPT-Vision (Chen et al., 2024a), have demonstrated the efficacy of adapting general-purpose VLMs to the medical domain via supervised fine-tuning on curated image-text datasets (Hu et al., 2024; Zhang et al., 2023). More recent efforts such as Med-R1 (Lai et al., 2025) and MedVLM-R1 (Pan et al., 2025) have incorporated reinforcement learning to further improve the robustness and generalization of the models' reasoning capabilities. However, these advances predominantly focus on generating textual chains of thought, neglecting explicit visual interaction during the reasoning process. To the best of our knowledge, our work is the first to endow medical VLMs with explicit visual reasoning capabilities, which we define as the ability to agentically interleave textual deliberation with executable image-manipulation actions. While powerful general models like GPT-4o possess strong visual understanding, they do not exhibit this interactive, tool-augmented reasoning loop. MedVR models a process analogous to how clinicians progressively refine their analysis through active visual inspection.This design promotes more interpretable and verifiable decision-making, aligning the model's behavior more closely with established clinical workflows.

**General-Purpose Visual Reasoning.**   Concurrently, research in the general domain has explored augmenting VLMs with capabilities for active perception. Works such as DeepEyes (Zheng et al., 2025), Pixel-Reasoner (Su et al., 2025) and Chain-of-Focus (Zhang et al., 2025b) enable iterative visual operations like zooming and region-of-interest selection to refine evidence gathering on natural images. However, their direct translation into the medical domain is impeded by the inherent complexity of clinical images, which exhibit subtle pathologies, modality-specific artifacts, and diverse object scales (Xu et al., 2025). Furthermore, these methods typically presuppose the availability of fine-grained grounding annotations (*e.g.*, bounding boxes) for cold start. This prerequisite is often infeasible in the medical field due to the high cost and specialized expertise required for annotation (Liu et al., 2025b). MedVR directly confronts this limitation. By obviating the need for any explicit grounding supervision, our framework not only alleviates the prohibitive annotation burden but also significantly advances the clinical viability of interactive medical VLMs.

## 3 METHODOLOGY

### 3.1 PRELIMINARIES

**Training Objective.**   The training of MedVR is cast as a policy optimization problem within a reinforcement learning (RL) framework. The VLM acts as an agent governed by a policy $\pi_\theta$, which is optimized to maximize the expected cumulative reward obtained from the generated reasoning trajectories. The objective is formally defined as:

$$\max_\theta \mathbb{E}_{(Q,I)\sim\mathcal{D},\mathcal{T}\sim\pi_\theta} \left\{ R(\mathcal{T}) - \beta D_{\text{KL}} \left[ \pi_\theta(\cdot|Q,I) \,\|\, \pi_{\text{ref}}(\cdot|Q,I) \right] \right\} \tag{1}$$

Here, $(Q, I)$ denotes a question-image pair from the dataset $\mathcal{D}$, and $\mathcal{T}$ represents a complete reasoning trajectory sampled from the policy $\pi_\theta$. $R(\mathcal{T})$ is the terminal reward for the trajectory, and the KL-divergence term, scaled by $\beta$, penalizes drastic deviations from the reference policy $\pi_{\text{ref}}$.

**Rollout Formulation.** Departing from conventional models that generate a monolithic textual output, our agent constructs a dynamic reasoning trajectory by meticulously interleaving textual deliberation with explicit visual operations. The agent's action space encompasses both the generation of tokens for its chain-of-thought and the invocation of specialized tool commands. In this work, we focus on a fundamental yet powerful visual operation: `Zoom-in`. When a `Zoom-in` command, including

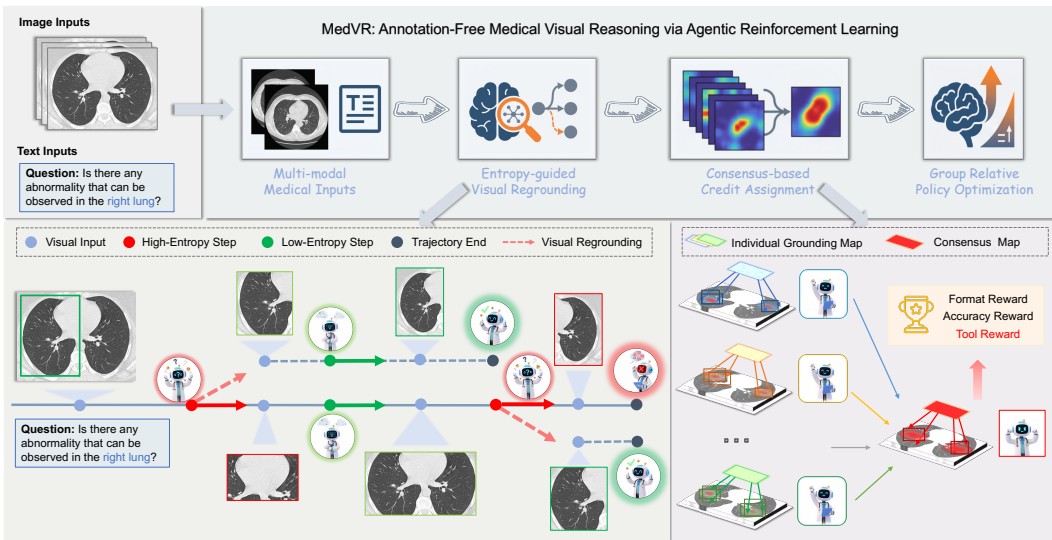

Figure 2: **Overview of MedVR.** The framework employs EVR to explore visual actions based on the model's intrinsic uncertainty and CCA to create a consensus-based reward from successful trajectories, enabling annotation-free training of medical visual reasoning.

its precise coordinates, is fully generated, an external tool is triggered to execute the corresponding action (*i.e.*, cropping a specified image region). The newly acquired visual information is then encoded into a set of special tokens and integrated back into the agent's context. This grounded visual evidence conditions all subsequent reasoning and action steps. This iterative process of deliberation, action, and observation continues until the agent produces a final answer and terminates the rollout.

**Reward Design.** To guide the agent's learning in the absence of fine-grained, step-by-step supervision, we employ a composite terminal reward function $R(\mathcal{T})$. This function is designed to provide a holistic evaluation of the entire trajectory and is defined as:

$$R(\mathcal{T}) = R_{\text{acc}}(\mathcal{T}) + R_{\text{format}}(\mathcal{T}) + \mathbb{1}(R_{\text{acc}}(\mathcal{T}) > 0) \cdot R_{\text{tool}}(\mathcal{T}) \tag{2}$$

The total reward comprises three components: a primary accuracy reward ($R_{\text{acc}}$) based on the correctness of the final answer; a minor format penalty ($R_{\text{format}}$) for syntactically invalid or malformed outputs; and a crucial tool-use reward ($R_{\text{tool}}$). The indicator function $\mathbb{1}(\cdot)$ means that the tool reward is conferred *only if* the trajectory successfully leads to a correct answer. This conditional structure is vital for incentivizing the model to discover meaningful causal relationships between its visual actions and successful outcomes, thereby discouraging spurious or ineffective tool use.

## 3.2 ENTROPY-GUIDED VISUAL REGROUNDING

A formidable challenge in training medical VLMs for intricate diagnostic tasks lies in guiding the agent to perform efficient and effective visual exploration. Without explicit supervision, an agent may struggle to learn *where* to look, often resorting to random or repetitive visual actions that are computationally expensive and yield little information. To overcome this, we introduce Entropy-guided Visual Regrounding (EVR), a novel exploration strategy that leverages the model's intrinsic uncertainty as a self-supervisory signal to dynamically guide the visual search process.

The fundamental premise of EVR is that moments of elevated predictive uncertainty during tool command generation correspond to critical junctures where the model lacks sufficient confidence in its impending visual action. When generating a `Zoom-in` command, an increase in entropy during the specification of coordinate tokens may indicate that while the model has identified the necessity for visual inspection, it remains uncertain about the precise region-of-interest (ROI). Rather than committing prematurely to a single, potentially suboptimal action, EVR harnesses this dynamic uncertainty signal to trigger a parallel exploration of multiple visual hypotheses. This mechanism provides an inherent, label-free form of supervision for the visual operation itself, compelling the model to explore diverse search strategies precisely at moments of heightened predictive uncertainty.

**Uncertainty Evaluation.** To operationalize this principle, we continuously monitor the token-level entropy of the policy $\pi_\theta$ as it parameterizes a visual action. The uncertainty at each generation step $t$ is quantitatively measured by the entropy of the next-token probability distribution:

$$H_t = -\sum_{j=1}^{|\mathcal{V}|} p_{t,j} \log p_{t,j}, \quad \text{where} \quad p_t = \pi_\theta(\cdot | S_{<t}) = \text{Softmax}\left(\frac{z_t}{T}\right) \tag{3}$$

Here, $S_{<t}$ denotes the sequence of previously generated tokens, $\mathcal{V}$ is the model's vocabulary, $z_t \in \mathbb{R}^{|\mathcal{V}|}$ are the pre-softmax logits, and $T$ is the decoding temperature. At the start of generation, we compute a baseline entropy $H_{\text{base}}$ as the mean entropy over the initial tokens, capturing the model's baseline uncertainty. Subsequently, during the generation of tool-relevant tokens, we continuously compute the rolling mean entropy $H_{\text{tool}}$ over the window. The key signal we track is the entropy increase relative to the baseline: $\Delta H_{\text{tool}} = H_{\text{tool}} - H_{\text{base}}$. A significant positive $\Delta H_{\text{tool}}$ serves as a robust indicator of escalating spatial uncertainty, signifying diminishing confidence in selecting a specific ROI and signaling an opportune moment to trigger an exploratory action.

**Adaptive Branching.** This dynamic uncertainty signal governs our exploration strategy through an adaptive branching mechanism. For each input, the total rollout budget is divided equally: half of the trajectories are generated initially to form a base set, while the remaining half are reserved for targeted exploration. Here, we define the branch probability $P = P_{\text{base}} + \gamma \Delta H_{\text{tool}}$, where $P_{\text{base}}$ is a base sampling probability to ensure a minimum level of exploration and $\gamma$ is the entropy weight that controls the influence of the measured entropy increase on the branching decision (Dong et al., 2025). During the generation of the base trajectories, while a tool's parameters are being decoded, EVR initiates a branching event stochastically with probability $P$. This process involves forking the current generative state and expending one unit from the discretionary budget to initiate a new, parallel reasoning path from this specific point of high uncertainty. Consequently, different branches can sample distinct coordinate values for the visual tool, effectively exploring multiple visual actions stemming from a shared reasoning context. This ensures that the exploration budget is strategically allocated to resolve the agent's spatial uncertainty at critical moments. The output of the EVR stage is a heterogeneous ensemble of $M$ trajectories, which collectively embody a diverse set of visual search hypotheses rooted in the model's own predictive uncertainty. This targeted collection of visual explorations provides a suitable foundation for our subsequent Consensus-based Credit Assignment (CCA) mechanism, designed to distill a robust supervisory signal from the ensemble.

### 3.3 CONSENSUS-BASED CREDIT ASSIGNMENT

While EVR generates diverse exploratory trajectories, the fundamental challenge of credit assignment remains: *How can we reward beneficial intermediate visual actions without ground-truth spatial annotations?* A singular reward for the final answer is inherently insufficient as it fails to distinguish between coincidental successes and inferences derived from sound visual grounding. To address this, we introduce Consensus-based Credit Assignment (CCA), a novel mechanism that provides fine-grained, self-generated supervision for visual operations.

The core principle of CCA is to harness the "wisdom of the crowd" by synthesizing information from the entire set of successful trajectories generated by EVR. The underlying assumption is that if multiple, diverse reasoning paths converge on a correct answer while consistently inspecting a particular image region, then that specific region is highly likely to be causally relevant to the solution. By identifying this consensus, CCA distills a high-confidence, dynamic pseudo-label for visual grounding, which in turn provides the basis for a nuanced grounding reward.

**Consensus Generation.** The CCA process begins by identifying the subset of successful trajectories, $\mathcal{T}^+$, from the ensemble produced by EVR. For each successful trajectory $\mathcal{T}_i \in \mathcal{T}^+$, we define its visual footprint as the union of all bounding boxes from its invoked `Zoom-in` operations. This footprint is then rasterized into a binary mask $M_i$. These individual masks are aggregated to form a consensus heatmap $C$, where the value at each pixel represents its inspection frequency across the successful set. This heatmap is subsequently binarized via a majority vote rule to yield the final consensus mask $\hat{M}$, which isolates regions inspected by a majority of successful trajectories:

$$\hat{M}(u, v) = \mathbb{1}(C(u, v) > |\mathcal{T}^+|/2), \quad \text{where} \quad C = \sum_{\mathcal{T}_i \in \mathcal{T}^+} M_i \tag{4}$$

where $\mathbb{1}(\cdot)$ is the indicator function. The resulting mask $\hat{M}$ represents a high-confidence hypothesis of the salient image region, derived entirely from the model's own successful explorations.

**Credit Assignment.** Utilizing the generated consensus mask $\hat{M}$, we formulate the tool reward $R_{\text{tool}}$ to explicitly encourage alignment with this collectively identified visual strategy. For each successful trajectory $\mathcal{T}_j \in \mathcal{T}^+$, we quantify its alignment with the consensus by computing the Intersection over Union (IoU) between its visual footprint $M_j$ and the consensus mask $\hat{M}$. To create a robust learning signal and mitigate noise inherent in pseudo-label generation, we employ a threshold $\eta$ on the IoU score. The final $R_{\text{tool}}$ is defined as:

$$R_{\text{tool}}(\mathcal{T}_j) = \begin{cases} 1.0, & \text{if} \quad \text{IoU}(M_j, \hat{M}) > \eta \\ 0.5, & \text{otherwise} \end{cases} \quad \text{where} \quad \text{IoU}(M_j, \hat{M}) = \frac{|M_j \cap \hat{M}|}{|M_j \cup \hat{M}|} \quad (5)$$

This tiered structure provides a baseline reward of 0.5 for achieving a correct outcome, while offering a significant bonus of 0.5 for doing so via a visual strategy that is consistent with the collective wisdom of other successful paths. This formulation rewards trajectories not merely for their final accuracy, but more importantly, for the verifiability and consistency of their underlying visual process.

The synergy between EVR and CCA thus establishes a fully self-supervisory learning loop. EVR acts as the *a priori* explorer, dynamically generating diverse visual hypotheses by focusing on moments of model uncertainty. Subsequently, CCA serves as the *a posteriori* distiller, extracting the collective wisdom from successful explorations to forge a fine-grained, self-generated reward signal. This symbiotic process guides the agent towards developing robust and interpretable visual reasoning capabilities, entirely obviating the need for costly, human-provided intermediate supervision.

## 4 EXPERIMENTS

### 4.1 EXPERIMENTAL SETUP

**Evaluation Benchmarks** We evaluate the effectiveness of MedVR across six public medical VQA benchmarks. Following established conventions (Huang et al., 2025), we categorize these benchmarks into two types: (1) General-Domain Medical VQA: This category includes OmniMedVQA (Hu et al., 2024), PMC-VQA (Zhang et al., 2023), and MedXpertQA (Zuo et al., 2025) datasets. These benchmarks encompass a diverse range of modalities and medical scenarios, with all questions presented in a **multiple-choice** format. (2) Modality-Specific Medical VQA: This category comprises benchmarks focusing on a single imaging modality, including VQA-RAD (Lau et al., 2018) (radiology X-rays), SLAKE (Liu et al., 2021) (slit-lamp ophthalmology images), and PathVQA (He et al., 2020) (pathology images). These benchmarks require open-ended, **free-text** answers.

**Evaluation Protocol.** For the general-domain multiple-choice tasks, we employ OmniMedVQA for training and in-domain testing, strictly adhering to the official data splits from Med-R1 (Lai et al., 2025). To assess out-of-domain (OOD) generalization, we conduct zero-shot evaluations on the test sets of PMC-VQA and MedXpertQA. For the modality-specific free-text tasks, the model is trained on a consolidated dataset of the VQA-RAD and SLAKE training splits, which constitute our in-domain benchmarks. We then evaluate its OOD generalization on PathVQA. Additional details regarding dataset statistics, evaluation metrics and baselines are provided in Appendix F and A.1.

**Implementation Details.** We implement MedVR using the Qwen2.5-VL-7B model as our backbone. The policy is optimized using the Group Relative Policy Optimization (GRPO) algorithm (Shao et al., 2024; Guo et al., 2025) for 64 iterations on a distributed setup of 32 H20 GPUs. During training, we use a batch size of 256 prompts. For each prompt, the agent generates 16 trajectories, with a maximum of 6 tool calls permitted. Key hyperparameters are set as follows: $P_{\text{base}} = 0.5$, $\gamma = 0.5$ for EVR, and $\eta = 0.5$ for CCA. In line with recent advancements in RLVR, we adopt the "clip-higher" mechanism (Yu et al., 2025) and ablate the KL-divergence and standard value normalization terms from the policy objective to stabilize training (Liu et al., 2025c).

### 4.2 MAIN RESULTS

As presented in Table 1, MedVR establishes a new state-of-the-art (SOTA) across a comprehensive suite of medical VQA benchmarks, outperforming both general-purpose and domain-specific vision-language models. The superiority of MedVR can be critically analyzed from three pivotal perspectives: **(1) Versatility Across Diverse Task Formats.** MedVR demonstrates remarkable adaptability by

Table 1: Comprehensive performance comparison on medical VQA benchmarks, divided into General-Domain (multiple-choice) and Modality-Specific (free-text) tasks. Out-of-domain (OOD) test sets are marked with $^\diamond$. The best and second-best results are highlighted in **bold** and underlined.

| Settings | General-Domain VQA (Multi-choice) | | | | Modality-Specific VQA (Free-text) | | | |
|---|---|---|---|---|---|---|---|---|
| Tasks | OMVQA | PMC-VQA$^\diamond$ | MedXQA$^\diamond$ | Avg. | VQA-RAD | SLAKE | PathVQA$^\diamond$ | Avg. |
| *General-Purpose VLMs* | | | | | | | | |
| Qwen2.5-VL-7B (Bai et al., 2025a) | 59.0 | 51.2 | 22.3 | 44.2 | 64.5 | 67.2 | 44.1 | 58.6 |
| Qwen2.5-VL-32B (Bai et al., 2025a) | 69.9 | 54.1 | 24.5 | 49.5 | 71.8 | 71.2 | 41.9 | 61.6 |
| InternVL2.5-8B (Chen et al., 2024c) | 83.9 | 51.3 | 21.0 | 52.1 | 59.4 | 69.0 | 42.1 | 56.8 |
| InternVL3-8B (Zhu et al., 2025) | 81.4 | 53.8 | 22.1 | 52.4 | 65.4 | 72.8 | 48.6 | 62.3 |
| InternVL3-14B (Zhu et al., 2025) | 81.9 | 54.1 | 23.1 | 53.0 | 66.3 | 72.8 | 48.0 | 62.4 |
| *Medical-Specific VLMs* | | | | | | | | |
| Med-R1-2B (Lai et al., 2025) | 77.3 | 47.4 | 21.1 | 48.6 | 39.0 | 54.5 | 15.3 | 36.3 |
| MedVLM-R1-2B (Pan et al., 2025) | 57.4 | 47.5 | 20.1 | 41.7 | 48.6 | 56.0 | 32.5 | 45.7 |
| MedGemma-4B (Sellergren et al., 2025) | 70.5 | 49.9 | 15.4 | 45.3 | 72.5 | 76.4 | 48.8 | 65.9 |
| LLaVA-Med-7B (Li et al., 2023a) | 29.3 | 30.5 | 20.3 | 26.7 | 53.7 | 48.0 | 38.8 | 46.8 |
| HuatuoGPT-V-7B (Chen et al., 2024b) | 80.1 | 53.3 | 22.3 | 51.2 | 67.0 | 67.8 | 48.0 | 61.6 |
| Lingshu-7B (Xu et al., 2025) | 84.2 | **54.3** | **26.5** | 55.0 | 67.9 | 83.1 | 61.9 | 70.3 |
| **MedVR (Ours)** | **96.8** | **54.3** | 26.4 | **59.2** | **74.4** | **85.3** | **62.3** | **74.0** |

excelling in both multiple-choice and free-text generation tasks. It not only achieves the highest score on the in-domain multiple-choice benchmark but also consistently outperforms all baselines on the more challenging free-text generation tasks. This consistent, top-tier performance across disparate answer formats indicates that the reasoning capabilities cultivated by MedVR are fundamental and generalizable, rather than being overfitted to a specific question-answering paradigm. This suggests our model learns to reason about visual content, not just to select the most probable option. **(2) Exceptional Out-of-Domain Generalization.** A critical indicator of a model's robustness is its ability to generalize to unseen data distributions. MedVR exhibits outstanding OOD performance, achieving SOTA or highly competitive results on all designated OOD test sets. This strong generalization capability suggests that our annotation-free training mechanisms effectively guide the model to develop a core, transferable reasoning skill grounded in visual evidence, thereby mitigating the risk of overfitting to dataset-specific biases and priors. **(3) Performance Rivaling Massive-Scale Pre-training.** A noteworthy achievement is that MedVR's performance is not only SOTA but also rivals or exceeds that of models pre-trained on substantially larger, domain-specific corpora. For example, MedVR consistently outperforms Lingshu-7B and the larger InternVL3-14B across the average scores for both task categories. This outcome is significant, as it demonstrates that targeted, principled improvements in reasoning processes can be more impactful than simply scaling up pre-training data. It underscores the efficacy and efficiency of our approach, which achieves superior results by fostering a deeper, evidence-based reasoning process rather than relying solely on the statistical patterns learned from massive datasets.

## 4.3 ABLATION STUDY

**Main Components of MedVR.** To dissect the contributions of each component within MedVR, we conduct a systematic ablation study. Starting from a textual RL baseline (Guo et al., 2025), we incrementally introduce our three core contributions (1) the `Zoom-in` visual tool, (2) the EVR exploration strategy, and (3) the CCA reward mechanism. The results are presented in Table 2.

First, we equip the agent with the `Zoom-in` tool without the guidance of EVR or CCA, which leads to a slight performance drop on the OmniMedVQA and PMC-VQA. This outcome suggests that VLMs do not inherently possess the zero-shot capability to effectively utilize novel tools. Without a sophisticated reward signal or exploration strategy, the tool can introduce unproductive search paths, disrupting the model's pre-established reasoning processes. Next, we isolate the individual impacts of EVR and CCA by adding them separately to the `Zoom-in` baseline. Integrating EVR alone yields substantial improvements, with the most significant gains on OOD benchmarks. This highlights EVR's role in enhancing model robustness and generalization by steering the agent to resolve predictive uncertainty

Table 2: Ablation study for core components of MedVR.

| Methods | Components | | | Performance | | |
|---|---|---|---|---|---|---|
| | Zoom-in | EVR | CCA | OmniMedVQA | PMC-VQA | MedXpertQA |
| Textual RL | — | — | — | 94.50 | 53.40 | 21.38 |
| MedVR | ✓ | — | — | 94.31 | 52.62 | 22.26 |
| | ✓ | ✓ | — | 95.38 | 53.81 | 24.73 |
| | ✓ | — | ✓ | 96.55 | 53.30 | 23.09 |
| | ✓ | ✓ | ✓ | **96.77** | **54.31** | **26.38** |

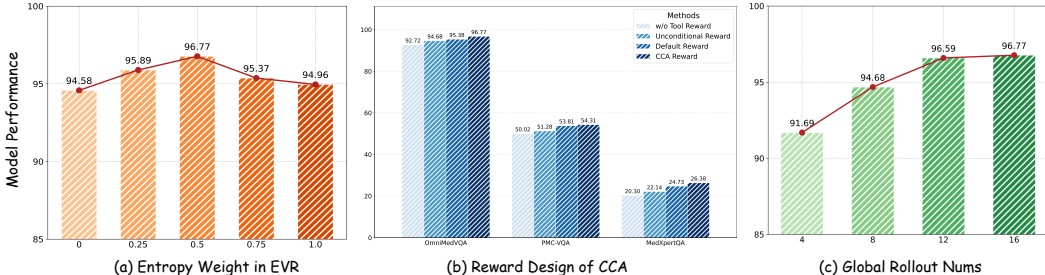

Figure 3: Ablation study for key hyperparameters and reward design of MedVR.

through visual exploration. Conversely, adding only CCA provides the most substantial benefit on the in-domain OmniMedVQA dataset, confirming its effectiveness at distilling and reinforcing reliable visual grounding strategies. Finally, the full MedVR model achieves the best performance across all benchmarks. This result illuminates a powerful synergy: EVR acts as an uncertainty-aware explorer, proposing diverse visual hypotheses, while CCA functions as a meticulous self-supervisor, identifying and rewarding consensus-driven, high-quality reasoning paths. This symbiotic interplay establishes a virtuous cycle of exploration and credit assignment, which is instrumental in unlocking the model's capacity for complex visual reasoning and generalization.

**Sensitivity to Entropy Weight in EVR.** We analyze the sensitivity of the EVR module to its entropy weight hyperparameter, $\gamma$, which governs the influence of predictive entropy on the tool-sampling process. As depicted in Figure 3(a), we vary $\gamma$ and evaluate performance on OmniMedVQA. At $\gamma = 0$, EVR defaults to a random sampling strategy, failing to leverage the model's intrinsic uncertainty to guide exploration. As $\gamma$ increases, the model progressively prioritizes actions that reduce predictive entropy, leading to a monotonic performance improvement up to $\gamma = 0.5$. Beyond this point ($\gamma > 0.5$), performance begins to decline. We attribute this to an overly greedy strategy where an excessive reliance on the entropy signal stifles exploration diversity, preventing the model from discovering a broader range of effective reasoning paths. This analysis validates our hyperparameter choice and underscores the critical trade-off between exploitation and exploration managed by EVR.

**Analysis of CCA Reward Design.** We also dissect the reward design of CCA by comparing it against three simplified variants, with results shown in Figure 3(b). The ablations are: (1) *w/o Tool Reward*, which removes any explicit reward for tool use; (2) *Unconditional Reward*, which provides a constant positive reward for any tool use, irrespective of final task accuracy; and (3) *Default Reward*, which grants a fixed reward only if the tool-using trajectory leads to a correct final answer. The results reveal a clear hierarchy in reward efficacy. The *w/o Tool Reward* setting fails to incentivize any visual exploration. The *Unconditional Reward* encourages tool use but fails to distinguish between beneficial and detrimental actions, as the reward is decoupled from task success. The *Default Reward* marks an improvement by linking tool rewards to final accuracy, but it treats all successful trajectories equally, lacking the granularity to reward more precise visual grounding. In contrast, our CCA mechanism resolves this by assigning credit based on cross-trajectory consensus. This enables fine-grained supervision that rewards not only a correct outcome but also the most reliable and replicable reasoning path to achieve it, proving crucial for the model's superior performance.

**Scalability of the Overall Framework.** To evaluate the scalability of the MedVR framework, we investigated the impact of global rollout numbers on model performance. The experiment was conducted on the OmniMedVQA dataset, with results presented in Figure 3(c). A clear positive correlation is observed: as the number of rollouts increases, the model's accuracy consistently improves. This trend demonstrates the scalability of our approach, as a larger rollout budget provides the CCA mechanism with a richer sample of trajectories to distill more reliable pseudo-supervision. Furthermore, this consistent improvement highlights the framework's robustness and its ability to effectively leverage increased computational resources.

## 4.4 IN-DEPTH ANALYSIS OF EVR

The fundamental premise of EVR is that the model's predictive uncertainty during the parameterization of a visual tool-call serves as a reliable signal to guide exploration. Rather than triggering exploration randomly, EVR directs the agent to explore alternative visual actions precisely at moments of high uncertainty, thereby optimizing the search for relevant visual evidence. We conducted several analyses to validate this hypothesis and demonstrate the multifaceted benefits of this approach.

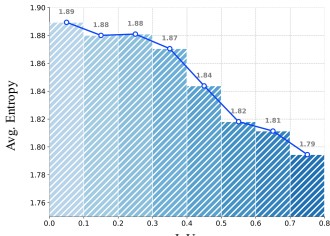 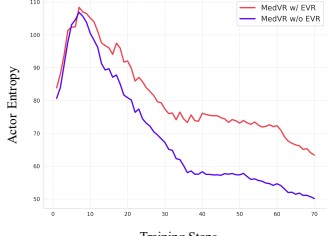 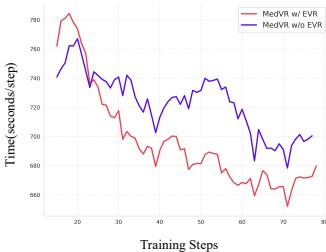

Figure 4: Correlation between token entropy and mIoU.

Figure 5: Average token entropy comparison with and w/o EVR.

Figure 6: Per-step training time comparison with and w/o EVR.

**Correlation between Entropy and Localization Quality.** We first sought to empirically validate our core assumption: that token-level entropy is a reliable proxy for the model's confidence in its visual grounding decision. To this end, we analyzed all tool-call tokens generated by Qwen2.5-VL-7B on the GEMEX-ThinkVG (Liu et al., 2025a) dataset, which provides grounding box annotations for visual references. We computed the average generation entropy of these tokens across different intervals of Intersection over Union (IoU) between the predicted and ground-truth bounding boxes.

As illustrated in Figure 4, we observe a strong negative correlation: high-quality localizations (high mIoU) consistently correspond to low-entropy generations, whereas high entropy is associated with inaccurate grounding. This finding confirms that entropy is a valid indicator of grounding uncertainty, justifying its use to trigger exploration precisely when a visual action is likely to be inaccurate.

**Impact on Training Stability and Exploration.** Beyond improving final performance, an effective exploration strategy should also promote stable learning dynamics. To evaluate this, we monitored the average policy entropy throughout the training process for models with and without EVR. As shown in Figure 5, the baseline model's entropy rapidly decays, indicating a premature convergence to a suboptimal, low-exploration policy. In contrast, EVR maintains a consistently higher and more stable level of entropy. This sustained exploration prevents the agent from getting stuck in local minima, facilitating the discovery of more robust reasoning paths and leading to better performance gains.

**Computational Efficiency of Tree-Search Exploration.** In addition to improving performance and stability, EVR's adaptive tree-search mechanism offers significant computational advantages over naive exploration strategies. By forking new reasoning paths from a shared context only at points of high uncertainty, EVR avoids the redundant token generation inherent in sampling multiple independent trajectories. Consequently, the computational complexity is potentially reduced from $O(n^2)$ to $O(n \log n)$. As quantified in Figure 6, EVR significantly reduces the computational overhead compared to the an independent sampling baseline. This efficiency renders sophisticated, targeted exploration computationally feasible.

## 4.5 GENERALIZATION ACROSS MODEL SCALES AND VARIANTS

To rigorously demonstrate that the contribution of our proposed framework is independent of model scale or specific variant, we conducted a series of controlled ablation studies across multiple backbone models: Qwen2.5-VL-3B, Qwen2.5-VL-7B, Qwen2.5-VL-32B, and Lingshu-7B on OmniMedVQA. For each backbone, we compared a zero-shot baseline, a standard Textual RL approach, and our MedVR framework, using the exact same RL settings, rollout budget, and training corpora.

As shown in Table 3, MedVR consistently and significantly outperforms the Textual RL baseline across all model sizes. These results demonstrate that the benefits of MedVR are not contingent on model size or a specific variant, but stem from our annotation-free visual reasoning method itself. Notably, the gains are most substantial in OOD settings, suggesting that visual rea-

Table 3: Performance of MedVR across model scales and variants.

| Model | Method | OmniMedVQA | PMC-VQA | MedXpertQA | Avg. |
|---|---|---|---|---|---|
| Qwen2.5-VL-3B | Zero-shot | 63.9 | **50.6** | 21.9 | 45.5 |
| | Textual RL | **93.8** | 47.5 | 20.2 | 53.8 |
| | **MedVR (Ours)** | 92.1 | 49.1 | **22.2** | **54.5** |
| Qwen2.5-VL-7B | Zero-shot | 64.3 | 51.2 | 22.3 | 46.0 |
| | Textual RL | 94.5 | 53.4 | 21.4 | 56.4 |
| | **MedVR (Ours)** | **96.7** | **54.3** | **26.4** | **59.1** |
| Qwen2.5-VL-32B | Zero-shot | 69.9 | 54.1 | 24.5 | 49.5 |
| | Textual RL | 95.9 | 53.2 | 25.2 | 58.1 |
| | **MedVR (Ours)** | **96.6** | **56.4** | **26.5** | **59.8** |
| Lingshu-7B | Zero-shot | 84.2 | 54.3 | 26.5 | 55.0 |
| | Textual RL | 95.7 | 53.7 | 25.1 | 58.3 |
| | **MedVR (Ours)** | **96.0** | **60.8** | **26.5** | **61.1** |

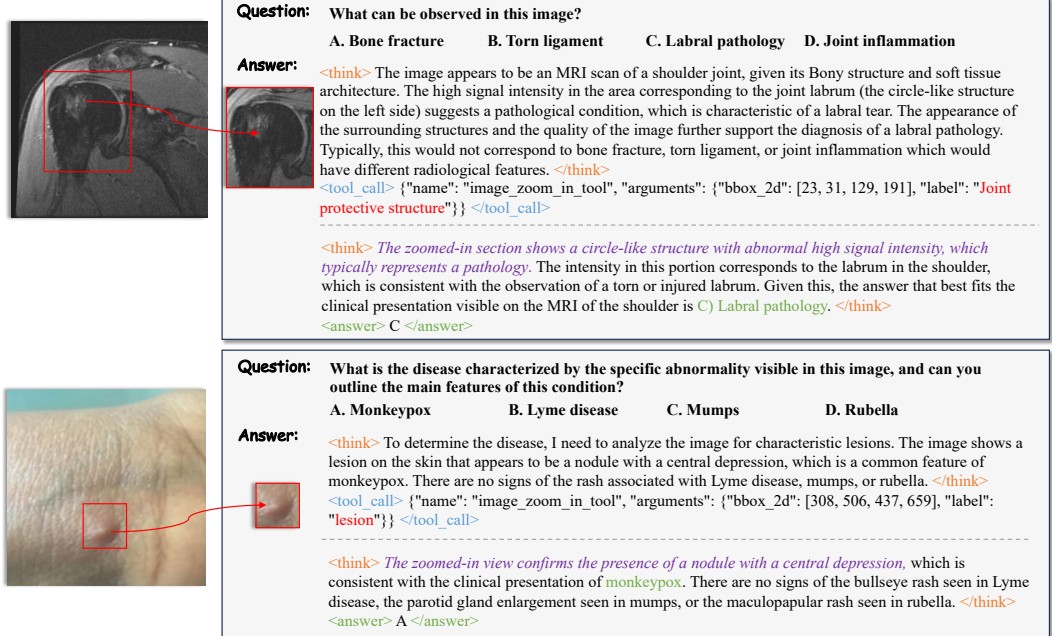

Figure 7: Qualitative examples of MedVR's visual reasoning process.

soning enhances robustness to domain shift by grounding decisions in observable image features rather than relying on memorized patterns. The consistent improvements across diverse backbones, ranging from general-purpose Qwen models to domain-specialized Lingshu, highlight the universality of MedVR's design. The framework is not tied to a specific architecture but can be readily integrated into any VLM capable of autoregressive policy generation and visual tool invocation.

## 4.6 CASE STUDY

To provide qualitative insights into MedVR's reasoning process, we present representative case studies in Figure 7. These examples showcase a dynamic, multi-step diagnostic workflow that mimics the iterative analysis of human clinical experts. As illustrated, the model typically initiates its analysis with a global overview of the image to formulate an initial hypothesis based on the user's query. Crucially, rather than prematurely arriving at a conclusion, MedVR identifies regions of uncertainty or high clinical relevance that necessitate closer inspection. At these critical junctures, it proactively invokes the Zoom-in tool for a targeted examination of the specific region of interest. The subsequent reasoning steps are then explicitly grounded in this newly acquired fine-grained visual evidence, leading to a final conclusion that is both accurate and visually verifiable. The visualizations confirm that MedVR's ability to localize and interpret clinically significant findings is highly precise.

This dynamic, iterative process stands in stark contrast to the static, single-pass analysis of conventional VLMs. By actively interrogating the visual data to resolve ambiguity and verify hypotheses, MedVR mitigates the risk of overlooking subtle pathologies. Furthermore, this grounding in direct visual evidence reduces the model's reliance on potentially unreliable textual priors, thereby curbing the tendency for hallucination. This capacity for self-correction and evidence-based reasoning is fundamental to enhancing the trustworthiness and reliability of AI in diagnostic applications.

## 5 CONCLUSION

This paper introduced MedVR, a novel reinforcement learning framework designed to overcome the critical limitations of text-only reasoning in medical VLMs. By seamlessly interleaving textual deliberation with image-tool use, MedVR grounds its inferences in visual evidence without relying on costly intermediate annotations. Our comprehensive experiments demonstrate that MedVR achieves state-of-the-art performance on diverse medical VQA benchmarks. The substantial gains over strong text-only baselines empirically confirm that learning to reason directly with visual data is essential for building robust and reliable medical AI.

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

APPENDIX

CONTENTS

# A    SUPPLEMENT EXPERIMENTAL RESULTS

To further validate the robustness, fairness, and generalizability of MedVR, we conducted a series of extended experiments. These include rigorously controlled comparisons against baselines, evaluations on diverse task formats and domains, and quantitative assessments of the model's core capabilities.

## A.1    BASELINE TRAINING DATA EXPOSURE AND FAIRNESS OF EVALUATION

A critical challenge in evaluating state-of-the-art vision-language models (VLMs) lies in ensuring fair and meaningful comparisons, particularly when benchmarking performance across models developed with heterogeneous training protocols, data sources, and architectural scales. Published baselines often differ significantly in their pretraining corpora, fine-tuning objectives, and exposure to evaluation benchmarks, which can introduce confounding variables that obscure the true contribution of methodological innovations.

To ensure transparency and provide a rigorous contextualization of MedVR's performance, especially in out-of-domain (OOD) settings, we present a comprehensive summary of the training data and learning paradigms used by leading medical VLMs in Table 4. This analysis reveals a crucial discrepancy: several strong baseline models were trained on datasets that substantially overlap with the benchmarks used for our OOD evaluation. For instance, Lingshu-7B and MedGemma-4B were both trained on a large-scale proprietary corpus that explicitly includes subsets of the OOD test sets, giving them an inherent advantage in generalization evaluations. In stark contrast, MedVR is trained exclusively on a filtered subset of *OmniMedVQA* with no access to any data from other benchmarks during training. This strict data isolation ensures a genuine out-of-distribution evaluation setup, making our model's competitive performance on these benchmarks a strong testament to its intrinsic generalization capability.

The data presented in Table 4 thus serves to clarify prior ambiguities in our initial manuscript, where the superiority of MedVR over models like Lingshu-7B on OOD tasks might have been misinterpreted. We emphasize that MedVR does not outperform Lingshu-7B due to data advantage, but rather demonstrates comparable or superior performance despite significantly less and less overlapping training data, underscoring the effectiveness of our annotation-free visual reasoning framework in inducing generalizable, data-efficient learning.

Table 4: Summary of training data, methodologies, and benchmark overlap for key baselines.

| Model | Training Data | Training Methods | Key Benchmarks Involved in Training |
|---|---|---|---|
| Med-R1-2B | OmniMedVQA | RL | OmniMedVQA |
| MedVLM-R1-2B | HuatuoGPT-Vision Eval | RL | OmniMedVQA, PMC-VQA, VQA-RAD, SLAKE, PathVQA |
| MedGemma-4B | Proprietary curated data (33M) | Pretraining, SFT, RL | PMC-VQA, VQA-RAD, SLAKE and others |
| LLaVA-Med-7B | PMC-15M | SFT | PMC-VQA |
| HuatuoGPT-V-7B | PubMedVision (1.3M) | Pretraining,SFT | PMC-VQA |
| Lingshu-7B | Proprietary curated data (12M) | Pretraining, SFT, RL | PMC-VQA, MedXpertQA, VQA-RAD, SLAKE, PathVQA and others |
| **MedVR (Ours)** | **OmniMedVQA (36K filtered)** | **RL** | **OmniMedVQA** |

## A.2    QUANTITATIVE EVALUATION OF LOCALIZATION QUALITY

A cornerstone of our work is the claim that MedVR learns to perform precise localization and robust reasoning without recourse to intermediate supervision. The trustworthiness of the entire framework hinges on the fidelity of these learned visual grounding steps. To provide direct, quantitative evidence for this claim, we conducted a rigorous evaluation of MedVR's localization capabilities on three diverse tasks that demand precise visual grounding:

1. **General Medical VQA** on the GEMEX-ThinkVG (Liu et al., 2025a) dataset, which requires locating anatomical structures or pathologies to answer questions.

2. **Medical Phrase Grounding** on the ChestX-ray8 (Wang et al., 2017) dataset, which tests the ability to map a textual phrase to a specific image region.

3. **Lesion Detection** on the ISIC (Codella et al., 2019) dataset, a classic task requiring precise outlining of skin lesions.

For each task, we measured the mean Intersection over Union (mIoU) between the bounding boxes generated by the model's `Zoom-in` tool and the ground-truth annotations.

The results, summarized in Table 5, reveal a dramatic improvement in localization accuracy for MedVR compared to the zero-shot Qwen2.5VL-7B backbone. The consistent, high-performance across these varied tasks demonstrates that the learned skill is not a narrow, task-specific artifact but a generalizable capability. The low standard deviation in our results further indicates stable and reliable performance. This provides strong empirical validation that our annotation-free learning process successfully distills a high-

Table 5: Quantitative evaluation of localization quality. MedVR significantly improves the mean Intersection over Union (mIoU) compared to the zero-shot backbone, demonstrating its ability to learn precise visual grounding without annotations.

| Model | GEMEX-ThinkVG | ChestX-ray8 | ISIC |
|---|---|---|---|
| Qwen2.5-VL-7B | 17.54 ± 2.13 | 36.53 ± 3.21 | 35.73 ± 1.87 |
| **MedVR (Ours)** | **59.62 ± 1.73** | **54.29 ± 1.81** | **69.12 ± 1.35** |

quality supervisory signal, effectively teaching the model to identify and focus on causally relevant clinical regions.

### A.3 EFFICACY OF CONSENSUS-BASED CREDIT ASSIGNMENT (CCA)

A critical question for any self-supervisory mechanism is whether it distills a true, meaningful signal or merely amplifies noise from the model's own biases. To validate that our annotation-free CCA mechanism generates high-quality pseudo-supervision, we compared its performance against a supervised upper bound. We conducted an experiment on the GEMEX-ThinkVG dataset, which provides ground-truth masks for relevant regions. In this oracle setting, the CCA consensus mask was replaced with the ground-truth mask, providing direct, perfect supervision for the `Zoom-in` tool reward.

The results are presented in Table 6. Strikingly, the performance of our fully annotation-free approach is highly comparable to that of its supervised counterpart, with only a marginal difference in both final task accuracy and localization mIoU. This result provides powerful evidence that CCA is highly effective at distilling a high-quality reward signal from the collective agreement of successful rollouts. It demonstrates that the "wisdom of the crowd" principle holds, suc-

Table 6: Comparison of MedVR with its supervised counterpart on GEMEX-ThinkVG.

| Method | Accuracy (%) | mIoU (%) |
|---|---|---|
| w/ Annotation | 79.62 | 61.33 |
| w/o Annotation | 79.08 | 59.62 |

cessfully guiding the agent toward semantically relevant regions without requiring any costly human annotations. This finding is central to the practical value of MedVR, as it confirms the viability of achieving robust visual grounding in a scalable, annotation-free manner.

### A.4 DOMAIN-AGNOSTIC GENERALIZATION TO NON-MEDICAL TASKS

To investigate whether our proposed mechanisms, EVR and CCA, constitute fundamental improvements in visual reasoning or are merely specialized heuristics for the medical domain, we assessed their efficacy on general-domain benchmarks. This evaluation serves to test the hypothesis that our contributions are domain-agnostic and broadly applicable. We evaluated our method on two categories of general-domain tasks: (1) **Visual Grounding**. For this task, our model was trained on the training splits of the RefCOCO/+/g (Chen et al., 2025) datasets and evaluated on the corresponding validation splits; and (2) **Multimodal Reasoning**. For this task, the model was first trained on Geometry3k (Lu et al., 2021) and then evaluated on three challenging benchmarks: MathVista (Lu et al., 2023), MathVerse (Zhang et al., 2024), and MathVision (Awais et al., 2024).

As detailed in Table 7a and Table 7b, our method consistently outperforms both the powerful Qwen2.5-VL-7B backbone and DeepEyes (Zheng et al., 2025), a strong contemporary model specialized in visual reasoning. The superior performance across these non-medical tasks provides conclusive evidence that EVR and CCA are not narrow, domain-specific solutions. Instead, they represent a

Table 7: Performance on visual grounding and multimodal reasoning benchmarks.

(a) Visual grounding benchmarks.

| Model | refCOCO | refCOCO+ | refCOCOg | Avg. |
|---|---|---|---|---|
| Qwen2.5VL-7B | 89.1 | 82.6 | 86.1 | 85.6 |
| DeepEyes | 89.8 | 83.6 | 86.7 | 86.7 |
| **EVR + CCA (Ours)** | **92.1** | **85.2** | **87.4** | **88.2** |

(b) Multimodal reasoning benchmarks.

| Model | MathVision | MathVerse | MathVista | Avg. |
|---|---|---|---|---|
| Qwen2.5VL-7B | 24.9 | 43.8 | 66.3 | 44.3 |
| DeepEyes | 26.6 | **47.3** | 70.1 | 47.9 |
| **EVR + CCA (Ours)** | **29.0** | 46.4 | **71.4** | **48.9** |

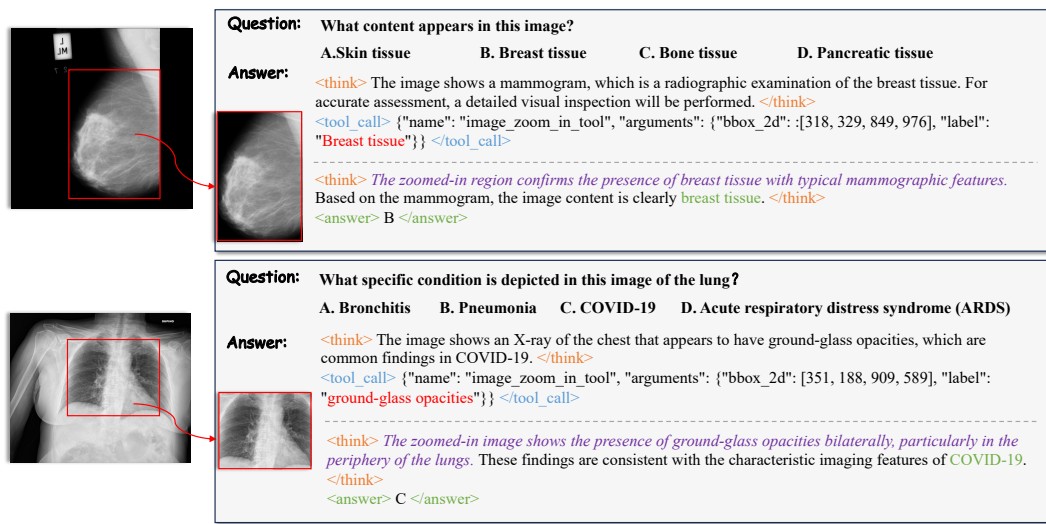

Figure 8: Additional Visualization Results of MedVR's Reasoning Process.

principled and generalizable approach to agentic visual reasoning. They effectively address the core challenges of exploration and credit assignment in a tool-augmented RL setting, regardless of whether the visual input is a chest X-ray or a complex geometric diagram. This confirms that MedVR's core contributions are of broad relevance to the wider field of multimodal AI.

## A.5 MORE VISUALIZATION RESULTS

We provide an extended collection of qualitative results to further illustrate the robustness and versatility of MedVR's visual reasoning capabilities. These examples, presented in Figure 8, supplement the case studies in Section 7 of the main paper and are chosen to demonstrate the model's performance across a wider range of medical modalities, question complexities, and clinical scenarios. The visualized cases have been selected to highlight several key strengths of our approach:

- **Cross-Modality Proficiency:** The examples span multiple imaging modalities, including chest X-rays, CT scans, and MRIs. This demonstrates that MedVR's learned reasoning strategy is not confined to a single type of medical image but generalizes effectively, correctly identifying regions of interest regardless of whether it's a skeletal structure in a radiograph or soft tissue in a magnetic resonance image.

- **Interleaved Textual and Visual Reasoning:** The provided cases compellingly demonstrate the model's capacity for a tight coupling of textual deliberation and visual action. This process is not a linear, one-shot analysis but rather a dynamic, interleaved reasoning chain. Initially, the model forms a textual hypothesis based on the global image and the query. This internal textual monologue then guides the decision to execute a visual action by invoking the `Zoom-in` tool on a region of interest. Subsequently, the new, high-resolution visual information is fed back into the reasoning process, allowing the model to update its textual understanding, refine its hypothesis, or even trigger further visual exploration.

- **Precise Localization and Grounding:** Across all examples, a consistent theme is the precision of MedVR's localization. The bounding boxes generated by the `Zoom-in` tool accurately frame the causally relevant anatomical or pathological features. The final

generated answer explicitly references the findings from this localized view, confirming that the model's conclusion is grounded in direct visual evidence rather than spurious correlations or textual priors. This is exemplified in the last case, where the model correctly identifies and describes a small, easily missed fracture.

Collectively, these additional visualizations reinforce the central thesis of our work: by equipping VLMs with a mechanism for active visual exploration and a robust self-supervisory signal for credit assignment, we can foster a more generalizable and trustworthy form of medical reasoning. The dynamic, evidence-seeking behavior demonstrated here stands in stark contrast to the static analysis of conventional models and represents a critical step towards building more reliable medical AI for clinical applications.

### A.6  Analysis of Learned Tool Usage Dynamics

To understand how the agent's behavior evolves during training, we analyzed the dynamics of tool usage over time. We tracked the average number of `Zoom-in` calls per trajectory throughout the training process, as depicted in Figure 9. The resulting curve reveals a classic learning pattern.

Initially, the number of tool calls increases as the agent explores the action space and discovers the utility of the tool for improving task success. Following this exploration phase, the number of calls gradually decreases and stabilizes at an efficient rate (approximately 1.05 calls per trajectory).

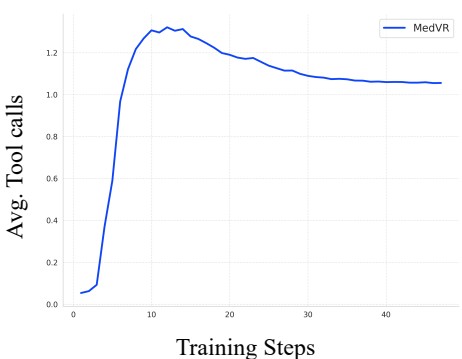

This trend demonstrates that the agent does not learn to call the tool randomly or excessively. Instead, through reinforcement learning, it masters a sophisticated and judicious policy for tool use. It learns to apply the `Zoom-in` function strategically, invoking it primarily when the expected information gain outweighs the operational cost. This convergence to an efficient and effective policy underscores the agent's ability to learn not just *how* to use a tool, but more importantly, *when* to use it to maximize its reasoning performance.

Figure 9: Average number of tool calls per trajectory during training.

## B  The Algorithm Workflow of MedVR

In this section, we provide a detailed flowchart of the MedVR algorithm in Algorithm 1.

## C  Supplement Related Work

This section provides an expanded discussion of related work to further contextualize the contributions of MedVR. We differentiate our framework from prior research in two key areas: annotation-free reinforcement learning and visual-grounded reasoning.

### C.1  Annotation-free Reinforcement Learning

Recent advancements in annotation-free reinforcement learning have demonstrated the potential of training language models using intrinsic rewards, obviating the need for human-annotated labels. Seminal works in this area include EMPO (Zhang et al., 2025a), which optimizes for semantic consistency by minimizing prediction entropy in a latent space; INTUITOR (Zhao et al., 2025), which leverages the model's self-certainty as an intrinsic reward signal; and MM-UPT (Wei et al., 2025), which employs a majority voting mechanism to generate pseudo-labels for reward calculation.

While MedVR draws inspiration from this paradigm of self-supervision, it addresses a fundamentally different set of challenges rooted in the unique demands of multimodal medical reasoning. Our contributions are distinguished from these prior works in several critical aspects:

---

**Algorithm 1** The MedVR Training Framework

---

1: **Input:** Initial policy model $\pi_{\theta_{init}}$; task dataset $\mathcal{D}$; hyperparameters for GRPO; EVR hyperparameters $P_{base}, \gamma$; CCA hyperparameter $\eta$; total rollouts per prompt $M$.

2: **Initialize:** Policy model $\pi_\theta \leftarrow \pi_{\theta_{init}}$.

3: **for** iteration $i = 1, \ldots, I$ **do**

4:      Freeze a reference policy $\pi_{ref} \leftarrow \pi_\theta$.

5:      **for** step $s = 1, \ldots, S$ **do**

6:          Sample a batch of prompts $\mathcal{D}_b = \{(Q_j, I_j)\}_{j=1}^B$ from $\mathcal{D}$.

7:                  ▷ — *Trajectory Generation with Entropy-guided Visual Regrounding (EVR)* —

8:          Initialize an empty set of trajectories for each prompt: $\mathcal{T}_j \leftarrow \emptyset$ for $j = 1, \ldots, B$.

9:          **Phase 1: Base Trajectory Generation & Adaptive Branching**

10:          Generate $M/2$ initial trajectories for each prompt by sampling from $\pi_{ref}$.

11:          **for all** each generated trajectory $\tau$ **do**

12:              Compute baseline entropy $H_{base}$ at the start of the trajectory.

13:              **if** a `<tool_call>` for `Zoom-in` is being generated **then**

14:                  Compute current tool-call entropy $H_{tool}$.

15:                  Calculate entropy increase $\Delta H_{tool} = H_{tool} - H_{base}$.

16:                  Define branch probability $P_{branch} = P_{base} + \gamma \Delta H_{tool}$.

17:                  **if** a random number $u \sim U(0, 1) < P_{branch}$ and exploration budget allows **then**

18:                      **Branch:** Fork the current generative state and generate a new parallel trajectory from this point of high uncertainty, adding it to the corresponding $\mathcal{T}_j$.

19:                  **end if**

20:              **end if**

21:          **end for**

22:          **Phase 2: Complete Remaining Trajectories**

23:          For each prompt, generate additional trajectories independently from $\pi_{ref}$ until $|\mathcal{T}_j| = M$.

24:                  ▷ — *Reward Calculation with Consensus-based Credit Assignment (CCA)* —

25:          **for all** prompt $j \in \{1, \ldots, B\}$ **do**

26:              Identify the set of successful trajectories $\mathcal{T}_j^+ = \{\tau \in \mathcal{T}_j \mid R_{acc}(\tau) > 0\}$.

27:              **if** $|\mathcal{T}_j^+| > 1$ **then**

28:                  Generate visual footprint masks $\{M_k\}$ for each $\tau_k \in \mathcal{T}_j^+$.

29:                  Aggregate masks to create a heatmap $C = \sum_k M_k$.

30:                  Binarize heatmap to obtain consensus mask $\hat{M}_j(u, v) = \mathbb{1}(C(u, v) > |\mathcal{T}_j^+|/2)$.

31:                  **for all** trajectory $\tau_k \in \mathcal{T}_j^+$ **do**

32:                      Compute $IoU(M_k, \hat{M}_j)$.

33:                      Assign consensus-aligned tool reward: $R_{tool}(\tau_k)$.

34:                  **end for**

35:              **end if**

36:              Compute total terminal reward $R(\tau)$ for all $\tau \in \mathcal{T}_j$ using Equation (2).

37:          **end for**

38:                        ▷ — *Policy Optimization* —

39:          Use the collected trajectories $\{\mathcal{T}_j\}_{j=1}^B$ and their associated rewards $\{R(\tau)\}$ to compute advantages.

40:          Update the policy model $\pi_\theta$ by maximizing the Group Relative Policy Optimization (GRPO) objective.

41:      **end for**

42: **end for**

43: **Output:** The optimized policy model $\pi_\theta$.

---

- **Problem Formulation and Definition of "Annotation-Free":** The primary distinction lies in the definition of what is "annotation-free." Prior works typically address tasks where the *final answer* is unavailable, optimizing for the internal consistency of the reasoning process itself. In contrast, our medical VQA setting provides ground-truth final answers. Our "annotation-free" challenge pertains to the intermediate reasoning steps. In a clinical workflow, a verifiable diagnostic process is as crucial as the final conclusion. However, annotating these fine-grained visual grounding steps is prohibitively expensive and requires specialized expertise. MedVR is designed specifically to learn this verifiable process without any intermediate supervision, a problem distinct from that addressed by prior text-only methods.

- **Research Domain and Modality:** The aforementioned methods focus almost exclusively on text-only reasoning and are primarily validated on tasks such as mathematics or commonsense reasoning. Our work tackles multimodal medical VQA, a domain where general-purpose Vision-Language Models (VLMs) face significant domain and modality gaps. Applying text-based intrinsic reward mechanisms directly is non-trivial and often insufficient for tasks that demand nuanced interpretation of fine-grained visual evidence, such as identifying subtle pathologies in medical images.

- **Agentic Framework vs. Single-Pass Generation:** Previous methods typically operate on a single-pass textual generation paradigm. MedVR, however, is built upon an agentic reinforcement learning framework that supports multi-turn, interleaved visual-textual reasoning. This allows the model to actively probe the visual input, refine its hypotheses, and ground its conclusions in a dynamic, iterative manner. To our knowledge, MedVR is the first work to develop an annotation-free optimization strategy specifically tailored for an agentic RL context in a complex multimodal setting.

- **Semantic and Interpretable Technical Approach:** Although we leverage concepts like entropy and voting, our implementations are fundamentally different and tailored for multimodal interpretability. Instead of optimizing at the token level, which lacks semantic completeness, our Entropy-guided Visual Regrounding (EVR) operates on the semantically complete tool_call level. Furthermore, our Consensus-based Credit Assignment (CCA) moves beyond simple textual voting; it distills a visual consensus mask from the grounding boxes generated across diverse rollouts, providing a highly interpretable supervisory signal directly within the visual domain.

In summary, MedVR represents a principled adaptation and extension of annotation-free learning, specifically engineered to solve the confluence of challenges inherent in building verifiable and trustworthy medical AI.

## C.2 Visual-Grounded Reasoning

The field of visual-grounded reasoning has seen significant progress, with models like ViGoRL (Sarch et al., 2025), UniVG-R1 (Bai et al., 2025b), COPL (Jiang & Lu, 2024), and VGR (Wang et al., 2025) demonstrating impressive capabilities in linking textual outputs to specific image regions. These methods have pushed the boundaries of how models can "see" and reason about the visual world.

Despite these advances, MedVR is distinguished by its unique objectives, data assumptions, and learning paradigm, which are specifically tailored to the constraints and requirements of the medical domain.

- **Research Objective: A Means, Not an End:** A key difference lies in the ultimate goal. Methods like UniVG-R1 and COPL primarily focus on visual grounding or localization as the end-goal itself. In contrast, MedVR leverages the model's grounding ability as an intermediate step within a broader diagnostic reasoning chain. For MedVR, precise localization is not the final output but a critical means to provide a verifiable and interpretable visual reasoning process that can aid clinical decision-making.

- **Data Dependency and the Annotation-Free Imperative:** The most critical distinction is the reliance on supervision. Prior works like ViGoRL and VGR critically depend on large-scale, human-annotated supervised fine-tuning (SFT) datasets that provide fine-grained grounding information to bootstrap the model's capabilities. This prerequisite is infeasible in

the medical domain, where such annotations are prohibitively expensive, require specialized clinical expertise, and are largely unavailable at scale. MedVR directly confronts this fundamental bottleneck. To our knowledge, it is the first framework to explore and achieve robust visual-grounded reasoning for medical VQA in a completely annotation-free manner, making it a scalable and practical solution for real-world medical AI development.

- **Methodological Focus on the RL Process:** The focus of many prior works, such as ViGoRL and UniVG-R1, is heavily weighted toward dataset construction and supervised fine-tuning. The reinforcement learning component, if present, is often not the central methodological innovation. MedVR, conversely, is the first work in this area to perform fine-grained modeling of the end-to-end agentic RL process for visual reasoning. Our novel mechanisms for uncertainty-guided exploration (EVR) and consensus-based reward shaping (CCA) represent a deep and principled exploration of the RL rollout and credit assignment problem in this context.

- **Learning Paradigm: Internal Refinement vs. External Supervision:** The underlying motivation of prior works is to enhance a model's grounding ability by learning from *external*, high-quality annotated data. This paradigm aims to "inject" grounding skill through SFT. In contrast, MedVR is designed to leverage and refine the model's *internal*, pre-existing grounding capabilities through a self-supervisory loop within the RL process itself. It teaches the model to improve its own innate skills, guided only by the final task reward and the consensus of its own successful explorations.

By obviating the need for intermediate annotations, MedVR offers a new paradigm for training visually-grounded medical AI that is both highly effective and scalable in the face of real-world data scarcity.

# D  IMPLEMENTATION DETAILS

## D.1  TRAINING INITIALIZATION

We clarify that our "annotation-free" RL training does not use any supervised fine-tuning (SFT) or instruction-tuning on human-annotated reasoning traces to teach the model its output format. The agent learns to produce structured outputs with thoughts and tool calls directly through reinforcement learning. This is achieved by providing a detailed system prompt (see Figure 4 in the main paper) to the Qwen2.5-VL-7B base model. The strong instruction-following capabilities of the backbone model allow it to adhere to the predefined JSON format for tool calls without requiring a separate SFT warm-up stage. The reasoning behaviors observed are therefore a direct result of the RL agent learning to operate within the constraints defined by the system prompt, guided by the task and tool rewards.

## D.2  TOOL IMPLEMENTATION

The `Zoom-in` tool is the primary mechanism through which the MedVR agent actively interrogates the visual input. Its implementation is designed to be both effective and computationally efficient.

**Zoom-in Tool Workflow.**    The workflow proceeds as follows:

1. **Invocation:** When the agent's policy generates a complete `Zoom-in` command containing valid bounding box coordinates, an external tool is triggered.

2. **Execution:** This tool operates on the **original, full-resolution input image** to ensure maximum fidelity. A high-resolution patch corresponding to the specified coordinates is cropped.

3. **Visual Encoding:** The cropped patch is subsequently processed by the *same* pre-trained vision encoder used for the full image. This generates a new set of visual tokens representing fine-grained information from the selected region of interest.

4. **Context Integration:** These new tokens are encapsulated within special markers (`<tool_response>` and `</tool_response>`) and integrated back into the agent's context sequence.

5. **Conditioned Generation:** The model's subsequent generation is thereby conditioned on both its prior reasoning chain and this new, targeted visual evidence. Crucially, as these observation tokens are part of the environment's response and not generated by the policy, they are masked out during the policy loss calculation.

**Image Resolution.** A distinction is made between training and inference resolutions to balance computational feasibility with performance. During training, to manage GPU memory constraints, input images with a longest side exceeding 1024 pixels are resized to a maximum of 1024 pixels while maintaining their aspect ratio; smaller images retain their original dimensions. During inference, however, the model operates on the **original, full-resolution images** to ensure that the `Zoom-in` tool can access the highest level of detail possible for precise analysis.

**Inference Latency.** To assess the practical impact of tool use on performance, we profiled the average inference latency on the OmniMedVQA dataset using H20 GPUs. As detailed in Table 8, the computational overhead introduced by the `Zoom-in` tool is negligible, constituting only 1.5% of the total inference time. This low overhead is attributable to the fact that the primary bottleneck in inference is typically the sequential, autoregressive generation of text. In contrast, the vision encoder forward pass is highly parallelizable and comparatively fast. Furthermore, the latency from tool execution can be further minimized through standard optimization techniques such as Key-Value (KV) Caching and parallelized tool execution, making the MedVR framework highly suitable for practical applications.

Table 8: Inference latency analysis for MedVR on the OmniMedVQA dataset.

| Model | Total Tokens | Total Latency | Extra Visual Tokens | Extra Latency |
|---|---|---|---|---|
| MedVR | 169.23 | 15.21s | 84.76 | 0.23s (~1.5%) |

## D.3 Evaluation Protocol and Metrics

For the general-domain multiple-choice tasks, we employ OmniMedVQA for training and in-domain testing, strictly adhering to the official data splits from Med-R1 (Lai et al., 2025). To assess out-of-domain (OOD) generalization, we conduct zero-shot evaluations on the test sets of PMC-VQA and MedXpertQA. For these benchmarks, we use accuracy as both the evaluation metric and the reinforcement learning reward signal. For the modality-specific free-text tasks, the model is trained on a consolidated dataset of the VQA-RAD and SLAKE training splits, which constitute our in-domain benchmarks. We then evaluate its OOD generalization on PathVQA. Evaluating open-ended text generation necessitates a more nuanced approach, particularly for designing the reward function. Following prior work (Rui et al., 2025), we formulate a composite reward signal, $R_{\text{open}}$, that balances lexical overlap with semantic similarity. It integrates n-gram-based metrics (BLEU-1 and ROUGE-1) with a semantic-based metric (BERTScore), defined as:

$$R_{\text{open}}(\mathcal{T}) = \frac{1}{2}\lambda \cdot (\text{BLEU}_1(\mathcal{T}) + \text{ROUGE}_1(\mathcal{T})) + (1 - \lambda) \cdot \text{BERTScore}(\mathcal{T}) \quad (6)$$

where $\lambda$ is a balancing hyperparameter and we set $\lambda = 0.8$. For the final evaluation of open-ended tasks, we employ the Qwen3-235B-A22B (Yang et al., 2025) to assess the correctness of the generated answers.

## E Inference Settings

To ensure the rigor, fairness, and reproducibility of our experimental results, we established a comprehensive and standardized inference protocol. This protocol governs all aspects of the evaluation process, from the initial prompt engineering to the technical parameters of the decoding process.

### E.1 Prompts

The agent's behavior is guided by a carefully designed set of prompts that define its operational context and task objectives. This process involves two key components: a System Prompt and a User Prompt.

```
System Prompt

You are an expert Medical Imaging AI Assistant.

# Tools
You may call one or more functions to assist with the user query. You are provided with function signatures within <tools></tools> XML tags:
<tools>
{
      "type":"function",
      "function": {
            "name":"image_zoom_in_tool",
            "description":"Zoom in on a specific region of a medical image by cropping it based on a bounding box (bbox) and an object label. Use this to inspect
                        potential abnormalities, lesions, or specific anatomical structures mentioned in the query.",
            "parameters": {
                  "type": "object",
                  "properties": {
                        "bbox_2d": {
                              "type": "array",
                              "items": {
                                    "type": "number"
                              },
                               "minItems": 4,
                              "maxItems": 4,
                              "description": "The bounding box of the region for detailed examination, as [x1, y1, x2, y2], where (x1, y1) is the top-left corner and (x2, y2)
                                          is the  bottom-right corner."
                        },
                        "label": {
                              "type": "string",
                              "description": "A brief medical label for the object in the bounding box."
                        }
                  }
            }
      }
}
</tools>

# Instructions
In your response, you need to first think about the reasoning process in the mind and then conduct function calling to get additional information if needed. The
reasoning process and function calling are enclosed within <think></think> and <tool_call></tool_call> tags respectively. The results of the function calls will be
given back to you after execution, and you can continue to call functions until you get the final answer for the user's question. Finally, if you have got the answer,
enclose it within <answer></answer> tags and do not continue to call functions.

For each function call, return a json object with function name and arguments within <tool_call></tool_call> XML tags:
<tool_call>
{"name": <function-name>, "arguments": <args-json-object>}
</tool_call>
```

Figure 10: System Prompt used in MedVR.

The **System Prompt**, detailed in Figure 10, is meticulously crafted to elicit precise, clinically-grounded visual reasoning within a strictly defined output format. It serves as the agent's core instruction set, defining the "rules of engagement" by specifying the available tools (i.e., the `Zoom-in` tool), their function signatures, and the required JSON format for invocation. Furthermore, it enforces a structured reasoning process that mirrors clinical diagnostic logic, compelling the model to interleave textual deliberation (`<think>`) with tool-based actions before arriving at a conclusion. This structure is essential for generating transparent and verifiable reasoning traces.

In stark contrast, the **User Prompt**, shown in Figure 11, is intentionally minimalist. It presents the question and without imposing any constraints or hints on the intermediate reasoning path. This simplicity ensures that the model's reasoning process is driven by its own learned policy and understanding of the task, rather than by explicit guidance in the prompt, thereby enabling a more robust and unbiased evaluation of its autonomous capabilities.

## E.2 INFERENCE PROTOCOL AND PARAMETERS

All inference experiments were conducted on a cluster of NVIDIA H20 GPUs, leveraging the vLLM inference engine for its high-throughput performance and efficient memory management via PagedAttention. The generation of each reasoning trajectory is an autoregressive process. We employed a nucleus sampling (top-p) decoding strategy to guide this process. The key parameters were set as follows: Temperature: $T = 0.0$, Top-p: $p = 1.0$. To ensure computational tractability and prevent degenerate, non-terminating reasoning loops, we imposed the following constraints:

- **Maximum Tool Calls:** A maximum of **4** tool calls were permitted per trajectory. This limit was determined to be sufficient for the complexity of the evaluated tasks without encouraging inefficient or repetitive actions.

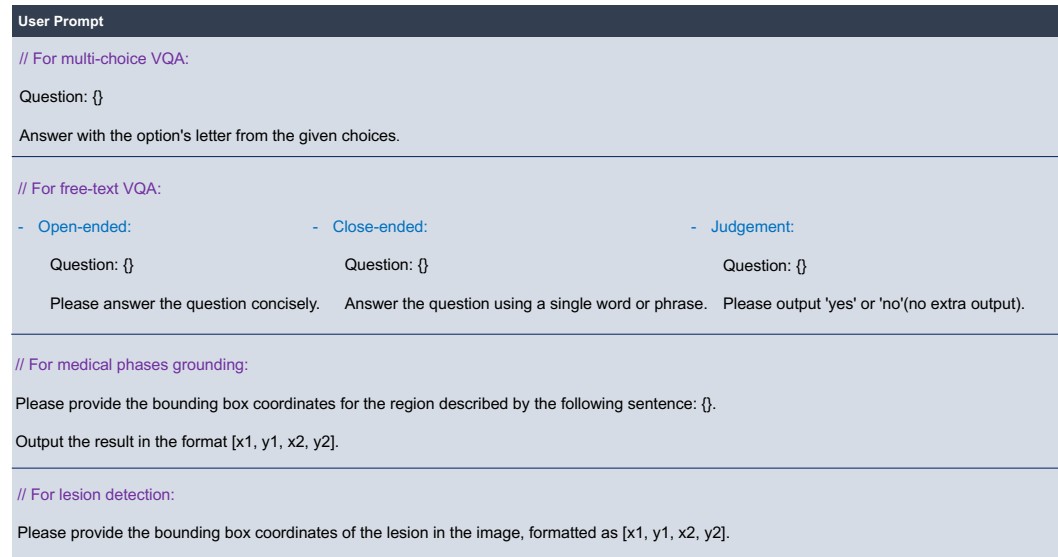

**User Prompt**

// For multi-choice VQA:

Question: {}

Answer with the option's letter from the given choices.

// For free-text VQA:

- Open-ended:           - Close-ended:           - Judgement:

    Question: {}            Question: {}            Question: {}

    Please answer the question concisely.     Answer the question using a single word or phrase.    Please output 'yes' or 'no'(no extra output).

// For medical phases grounding:

Please provide the bounding box coordinates for the region described by the following sentence: {}.

Output the result in the format [x1, y1, x2, y2].

// For lesion detection:

Please provide the bounding box coordinates of the lesion in the image, formatted as [x1, y1, x2, y2].

Figure 11: User Prompt for inference.

- **Maximum Token Length:** A global maximum token limit of **4096** was set for each turn. This serves as a safeguard against excessively long or runaway outputs.
- **Termination Signal:** The primary termination condition for a trajectory is that no tool call tokens are generated, which signals that the agent has concluded its reasoning process and is providing its final output.

# F DATASETS

**OmniMedVQA (Hu et al., 2024)** We utilize the publicly available subset of the OmniMedVQA benchmark as our in-domain dataset for medical visual question answering (VQA). This dataset comprises a total of 82,059 medical images and 88,996 image-question-answer triplets, spanning eight major imaging modalities: CT (15,808), MRI (31,877), X-Ray (7,916), Ultrasound (10,991), Dermoscopy (6,679), Fundus (5,398), OCT (4,646), and Microscopy (5,680). The questions are systematically categorized into five distinct types: Anatomy Identification (16,448 instances), Disease Diagnosis (55,387), Lesion Grading (2,098), Modality Recognition (11,565), and Other Biological Attributes (3,498), enabling comprehensive evaluation across diverse clinical reasoning tasks. To ensure consistency with prior work, we adopt the train-test split configuration used in Med-R1, facilitating direct comparison with existing models. In particular, to enhance training efficacy and concentrate on more challenging examples, we implement a data filtering strategy on the training set: we leverage Qwen2.5-VL-7B (Bai et al., 2025a) to generate 8 responses for each question and estimate the sample's difficulty based on the resulting accuracy. Samples where the model exhibits high confidence, defined as an accuracy of $\geq 7/8$, are subsequently excluded. This filtering process yields a curated, more challenging training dataset of 36,497 samples.

**PMC-VQA (Zhang et al., 2023)** PMC-VQA is a large-scale medical visual question answering dataset constructed through an automated pipeline that extracts and aligns multimodal content from open-access biomedical literature in the PubMed Central (PMC) archive. It contains 227,000 question-answer pairs associated with 149,000 unique medical images, covering a wide range of imaging modalities, anatomical regions, and pathological conditions. The dataset was designed to support generative MedVQA, where models are required to produce free-form textual answers rather than selecting from predefined options, thereby better simulating real-world clinician-AI interaction scenarios.

**MedXpertQA (Zuo et al., 2025)** MedXpertQA is an expert-level multimodal benchmark designed to assess advanced medical reasoning and domain-specific knowledge in artificial intelligence systems.

Comprising 4,460 carefully curated questions across 17 medical specialties and 11 body systems, MedXpertQA goes beyond conventional VQA datasets by incorporating complex clinical contexts, including patient histories, laboratory results, and multi-image analyses. Unlike datasets derived from automated captioning or simplified QA generation, MedXpertQA leverages authentic board-certification examination questions and undergoes rigorous synthesis and filtering procedures to prevent data leakage and ensure clinical validity.

**SLAKE (Liu et al., 2021)**    SLAKEis a large-scale bilingual Medical VQA dataset in English and Chinese, built upon multimodal radiology images—including CT, MRI, and X-ray, with rich semantic annotations, diverse QA pairs, and an integrated structured medical knowledge graph. It spans five anatomical regions: head, neck, chest, abdomen, and pelvic cavity, and supports both visual and knowledge-driven reasoning. The dataset aims to foster the development of medical AI capable of interpreting clinical images and providing accurate, context-aware answers to domain-specific questions.

**PathVQA (He et al., 2020)**    PathVQA is a specialized visual question answering (VQA) dataset designed for pathology, comprising 4,998 pathology images and 32,799 question-answer pairs. The images were sourced from two pathology textbooks and the PEIR Digital Library, and the QA pairs were generated via a semi-automated pipeline that extracts captions and transforms them into questions using natural language processing techniques, followed by manual verification to ensure accuracy. Unlike most medical VQA datasets that focus primarily on radiology (e.g., X-ray, CT, MRI), PathVQA centers on histopathological and cytological slides, along with some clinical photographs of pathological conditions, thereby addressing a distinct and underrepresented domain in medical AI. As the first publicly available VQA benchmark dedicated to pathology, PathVQA plays a pivotal role in advancing AI systems toward expert-level diagnostic capabilities in digital pathology

**VQA-RAD (Lau et al., 2018)**    VQA-RAD is a radiology-focused medical visual question answering (Med-VQA) dataset designed for training and evaluating AI systems that interpret clinical images and answer related questions. It comprises 2,244 unique question-answer pairs associated with 314 radiology images, sourced from the open-access MedPix database. The dataset includes both open-ended and binary questions, all of which were manually crafted by a team of clinicians to reflect authentic diagnostic inquiry patterns. Originally released with 2,248 question-answer pairs across 315 images, the dataset was later refined by removing four problematic entries—three duplicate triplets in the training set and one triplet shared between training and test sets—to ensure clean evaluation conditions. VQA-RAD serves as a foundational benchmark in Medical VQA research due to its clinical authenticity and balanced design.

# G    BASELINES

## G.1    GENERAL-PURPOSE VLMS

**Qwen2.5-VL (Bai et al., 2025a)**    Qwen2.5-VL extends the powerful Qwen2.5 language model by integrating sophisticated multimodal capabilities. Its vision encoder is significantly enhanced through techniques such as dynamic resolution training, window attention, SwiGLU, and RM-SNorm. This architecture enables Qwen2.5-VL to achieve robust visual reasoning performance across a diverse range of visual inputs, including charts, text documents, and complex layouts, making it a strong baseline for general multimodal tasks.

**InternVL2.5 (Chen et al., 2024c)**    InternVL2.5 is a prominent general-purpose VLM known for its strong performance in various multimodal benchmarks. As part of the InternVL series, it typically features a large-scale vision encoder meticulously aligned with a powerful large language model. This architecture allows it to process and understand complex visual information alongside textual prompts, demonstrating impressive capabilities in visual question answering, image captioning, and visual-grounded dialogue, thus representing a cutting-edge general VLM baseline.

**InternVL3 (Zhu et al., 2025)**    InternVL3 is a decoder-based multimodal model that combines a Qwen2.5-derived language backbone with a newly pre-trained vision encoder, following the ViT-MLP-LLM architecture. It is trained natively on text, images, and videos, enabling strong long-context

understanding and effective tool-use reasoning across diverse domains such as 3D vision and GUI agents.

## G.2 MEDICAL-SPECIFIC VLMS

**Med-R1-2B (Lai et al., 2025)** Med-R1-2B is a specialized reinforcement learning-enhanced vision-language model tailored for diverse medical tasks. A key innovation of Med-R1 is its employment of Group Relative Policy Optimization (GRPO) to significantly improve generalization capabilities in critical medical applications such as disease diagnosis and lesion grading. Notably, Med-R1-2B has demonstrated superior performance, surpassing larger general-purpose models like Qwen2-VL-72B by a significant margin on the OmniMedVQA dataset.

**MedVLM-R1-2B (Pan et al., 2025)** MedVLM-R1-2B stands as a highly specialized medical Vision-Language Model, meticulously engineered to achieve advanced visual and linguistic comprehension within the intricate domain of medical imaging and clinical documentation. It is also optimized through Group Relative Policy Optimization (GRPO). This sophisticated reinforcement learning approach enables MedVLM-R1-2B to learn robust reasoning strategies, specifically enhancing its ability to interpret complex medical imagery and generate medically pertinent and actionable responses.

**MedGemma-4B (Sellergren et al., 2025)** MedGemma-4B is a medical Vision-Language Model built upon the Gemma 3 language model. It integrates a specialized MedSigLIP encoder, which is pre-trained on a vast collection of de-identified medical images. The language components of MedGemma are further refined through training on extensive medical text corpora. Optimized for instruction tuning, MedGemma-4B is designed to support a range of medical applications, including the summarization of clinical reports and the generation of diagnosis explanations, thereby aiming to assist healthcare professionals.

**LLava-Med-7B (Li et al., 2023b)** LLaVA-Med-7B is a medical adaptation of the popular LLaVA framework, instantiated with a 7B parameter language model backbone. LLaVA models typically connect a pre-trained vision encoder to a large language model via a lightweight projection layer, and are trained on multimodal instruction-following data. LLaVA-Med-7B specializes in medical visual question answering and dialogue, having been fine-tuned on medical image-text datasets to imbue it with domain-specific knowledge and reasoning capabilities crucial for clinical tasks.

**HuatuoGPT-V-7B (Chen et al., 2024a)** HuatuoGPT-V-7B is a multimodal extension of the HuatuoGPT series, originally designed as a Chinese medical instruction-tuned Large Language Model (LLM) based on LLaMA. While the provided reference specifically describes the textual LLM, HuatuoGPT-V-7B would integrate a vision component to extend its diagnostic consultation capabilities to multimodal inputs. The original HuatuoGPT was fine-tuned using synthetic instructions generated by ChatGPT and authentic doctor-patient dialogues, enabling strong performance in symptom interpretation and treatment recommendation. HuatuoGPT-V-7B aims to bring this domain expertise to medical image understanding.

**Lingshu-7B (Xu et al., 2025)** Lingshu-7B is a comprehensive medical Multimodal Large Language Model (MLLM) built upon the Qwen2.5-VL architecture. It features a robust integration of a vision encoder, a powerful LLM, and a projection module to facilitate multimodal understanding. Lingshu-7B undergoes a multi-stage training process, critically incorporating Reinforcement Learning (RL) with verifiable rewards. It leverages an extensive dataset comprising over 5 million multimodal and textual medical samples, enabling unified understanding across various medical imaging types. This sophisticated training paradigm aims to deliver high-performance medical reasoning.

## G.3 FINE-TUNED VLMS

**Textual RL** To provide a clear benchmark for the efficacy of our interleaved visual-textual reasoning, we implemented a Textual RL baseline. This model serves as a direct counterpart to MedVR, designed to answer the question: "How much performance is gained by adding explicit visual interaction?" To this end, we ensured maximum controlled consistency between the two models. The Textual

---

**System Prompt for Textual RL**

A conversation between User and Assistant. The user asks a question, and the Assistant solves it. The assistant first thinks about the reasoning process in the mind and then provides the user with the answer. The reasoning process and answer are enclosed within <think></think> and <answer></answer> tags, respectively, i.e., <think> reasoning process here </think> <answer> answer here </answer>.

Figure 12: System Prompt used in Textual RL.

---

**User Prompt for Textual RL**

Question: {}.
Your task:
1. Think through the question step by step, enclose your reasoning process in <think>…</think> tags.
2. Then provide the correct single-letter choice (A, B, C, D,...) inside <answer>…</answer> tags.
3. No extra information or text outside of these tags.

Figure 13: User Prompt used in Textual RL.

---

RL baseline was trained using the same optimization algorithm (GRPO), training dataset, and key hyperparameters (e.g., batch size) as MedVR. Advanced optimization strategies, such as the "clip-higher" mechanism, were also identically applied to both setups.

The fundamental divergence is in the model's action space and its corresponding guidance. We crafted prompts inspired by the text-only reasoning paradigm of Med-R1, as shown in Figure 12 and Figure 13. These prompts instruct the model to formulate its reasoning as a purely textual chain-of-thought, processing the image as a static input without any interactive capabilities. While the Textual RL model demonstrates strong performance in its own right, its reliance on textual deliberation alone proves to be a significant limitation, particularly on tasks requiring fine-grained visual analysis. Its performance, therefore, serves as a robust reference point that underscores the substantial improvements afforded by MedVR's ability to ground its reasoning in dynamic visual evidence.

## H    LIMITATIONS AND FUTURE WORKS

While MedVR demonstrates significant advances, we acknowledge several limitations that highlight promising directions for future research. First, the current framework relies solely on a `Zoom-in` tool. Future work should expand the action space to include more sophisticated clinical tools, such as measurement, contrast adjustment (windowing), and multi-image comparison, to address a broader range of complex diagnostic tasks like tumor growth tracking. Second, our evaluation primarily focuses on final-answer accuracy, indirectly assessing the quality of visual grounding. A crucial next step is to conduct formal evaluations with clinical experts to score the plausibility and clinical relevance of the model's intermediate reasoning steps, providing a more direct validation of our approach. Finally, the reinforcement learning process can be computationally intensive. Future research could investigate more sample-efficient methods, such as offline RL or curriculum learning, to improve training stability and scalability, facilitating the application of MedVR to even larger models and more diverse datasets.

## I    ETHICS STATEMENT

This research strictly adheres to ethical guidelines for the use of medical data. Our experiments were conducted exclusively on publicly available medical Visual Question Answering (VQA) benchmarks. All data utilized in these datasets are fully anonymized to safeguard patient confidentiality, and their use in research is compliant with their respective data usage agreements. No new patient data was collected for this study.

## J    REPRODUCIBILITY STATEMENT

We are committed to ensuring the reproducibility of our research. In the main body of the paper and this appendix, we have provided comprehensive details regarding our model architecture, training

protocols, and hyperparameter settings. The implementation of our core mechanisms, Entropy-guided Visual Regrounding (EVR) and Consensus-based Credit Assignment (CCA), is described in detail to facilitate replication. We plan to release the source code for MedVR and the necessary configuration files to the research community upon completion of our institution's internal review process, thereby allowing for full verification of our findings.

## K  THE STATEMENTS OF USING LARGE LANGUAGE MODELS

During the preparation of this paper, we utilized large language models (LLMs) as a writing aid. Their use was strictly confined to improving the clarity, grammar, and style of the manuscript. The core scientific contributions, including the conceptualization of the MedVR framework, the design of experiments, the analysis of results, and the final conclusions, are entirely the intellectual product of the authors. The LLMs did not generate any scientific content or influence the interpretation of our findings.

