# OpenReview forum: "MedVR: Annotation-Free Medical Visual Reasoning via Agentic Reinforcement Learning"
_ICLR.cc/2026/Conference — ICLR 2026 Poster_

### Official Review · Reviewer_9mPg · 2025-10-27

**Soundness:** 2
**Presentation:** 2
**Contribution:** 2
**Rating:** 2
**Confidence:** 4

**Summary:**

MedVR addresses this by framing the VLM as an agent that can interleave text generation with a `Zoom-in` tool to actively inspect image regions. To train this agent without supervision, the paper proposes two novel, annotation-free mechanisms:
1.  **Entropy-guided Visual Regrounding (EVR):** This mechanism monitors the model's predictive uncertainty (entropy) during the generation of tool commands. When uncertainty is high (e.g., the model is unsure *where* to zoom), EVR triggers exploratory branching to sample different visual actions.
2.  **Consensus-based Credit Assignment (CCA):** This mechanism provides a reward signal for visual grounding. It identifies the set of all *successful* reasoning trajectories (those that led to a correct final answer) and generates a "consensus mask" based on the image regions they collectively inspected. Trajectories that align with this consensus receive an additional reward.

The authors claim that this self-supervising loop of exploration (EVR) and credit assignment (CCA) enables MedVR to achieve state-of-the-art performance on several medical VQA benchmarks, improving both accuracy and robustness.

**Strengths:**

1.  The paper tackles a key problem in medical VLMs: the lack of verifiable visual grounding. The proposed "annotation-free" approach, which avoids the need for expensive, step-by-step human annotations, is highly significant.
2.   The two core contributions, EVR and CCA, are original. Using intrinsic model uncertainty (entropy) to guide visual exploration (EVR) is an intelligent way to focus the agent's attention. Distilling a reward signal from the "consensus" of successful trajectories (CCA) is a clever solution to the credit assignment problem in the absence of ground truth.
3.   The paper correctly identifies the limitations of text-only reasoning and frames the solution as an agentic VLM that can interact with the image, mimicking a human clinician's workflow (e.g., zooming).

**Weaknesses:**

1.  **Misrepresentation of Results:** The most severe weakness is the contradiction between the paper's text and its tables. The text in Section 4.2 claims to "surpass" a key baseline on PMC-VQA, but Table 1 shows a marginal improvement (54.31 for MedVR vs. 54.29 for the Lingshu). This misrepresentation of out-of-domain performance is unacceptable and destroys confidence in the paper's conclusions.
2.  **Critical Omission of Image Resolution:** The paper's methodology is centered on a `Zoom-in` tool. This tool's function is entirely dependent on the input image resolution, which is never stated. It is impossible to know if the agent is (a) meaningfully cropping a high-resolution image or (b) trivially cropping an already-low-resolution 336x336 patch. This is a fundamental flaw that makes the core premise of the paper unverifiable.
3.  **No Evaluation of Core Detection Capability:** The `Zoom-in` tool and the entire CCA reward mechanism are critically dependent on the model's ability to *correctly* localize objects of interest. The paper provides no evaluation of the model's baseline detection or localization performance. This is a crucial omission. If the model's inherent ability to draw bounding boxes is poor, then the EVR is exploring random locations and the CCA is finding a "consensus" among incorrect boxes, which would poison the reward signal. The paper must demonstrate that the agent's visual grounding capabilities are a reliable foundation for its reasoning.
4.  **Lack of Model Scaling Analysis:** The experiments are confined to a single 7B model. There is no analysis of how this method performs at different scales (e.g., on a 3B model or a 30B+ model). This makes it unclear if the approach is a general one or a finicky result specific to the 7B backbone architecture.
5.  **Marginal OOD Performance:** Even taking the results at face value (and ignoring the misrepresentation), the OOD performance is not a strong endorsement. The model fails to demonstrate a substantial improvement on PMC-VQA (even this is easier than MedXpertQA) and is effectively tied on MedXpertQA. Given that the paper's premise is a more robust, generalizable reasoning process, these OOD results are weak.

**Questions:**

1.  What is the input image resolution used for the VLM? How does the `Zoom-in` tool work? Does it crop the original high-resolution image, or does it simply crop the low-resolution feature map? This detail is critical for understanding if the tool is performing a meaningful action.
2.  Can you provide any quantitative evaluation of the backbone model's zero-shot detection or localization capabilities? How can we be sure that the `Zoom-in` actions, which are the foundation of your method, are targeting clinically relevant regions?
3.  Why were no experiments conducted at other model scales (e.g., 3B or 34B)? How can readers be confident that this complex RL framework is scalable and not just tuned to one specific 7B backbone model?
4.  Regarding the OOD baseline comparisons, were all baseline models trained *only* on the OmniMedVQA training split to ensure a fair, controlled comparison for generalization? Or were your training set has similar case with the OOD data?

---

> ### Author Response · Authors · 2025-11-27
> **Response to Reviewer 9mPg (1/3)**
>
> We sincerely thank the reviewer for the detailed and insightful feedback. The comments have been invaluable in helping us clarify our contributions and strengthen our paper. We have conducted several new experiments and revised the manuscript to address all the points raised. Below, we provide a point-by-point response.
>
> ---
>
> **Response to W1, W5, Q4: Misrepresentation, OOD Performance, and Fairness of Comparisons**
>
> We thank the reviewer for raising these critical points regarding the fairness of our baseline comparisons and the interpretation of our out-of-domain (OOD) results. We acknowledge that our initial presentation may have caused confusion, and we have now taken several steps to provide a clearer, more transparent, and fairer evaluation of MedVR.
>
> **1. Clarification on Baseline Training Data and Context for OOD Performance**
>
> **1.1 Baseline Training Data** We agree that direct comparisons with published baselines are challenging due to variations in training data and methodologies. To address this, we have added a comprehensive table to **Appendix  A.1** (and a summary below) detailing the training specifics of each baseline model.
> | Model | Training Data | Training Methods | Key Benchmarks Involved in Training |
> | --- | --- | --- | --- |
> | Med-R1-2B | OmniMedVQA | RL | OmniMedVQA |
> | MedVLM-R1-2B | HuatuoGPT-Vision Eval | RL | OmniMedVQA, PMC-VQA, VQA-RAD, SLAKE, PathVQA |
> | MedGemma-4B | Proprietary curated data (33M) | Pretraining, SFT, RL | PMC-VQA, VQA-RAD, SLAKE and others |
> | LLaVA-Med-7B | PMC-15M | SFT | PMC-VQA |
> | HuatuoGPT-V-7B | PubMedVision (1.3M) | Pretraining, SFT | PMC-VQA |
> | **Lingshu-7B** | **Proprietary curated data (12M)** | **Pretraining, SFT, RL** | **PMC-VQA, MedXpertQA, VQA-RAD, SLAKE, PathVQA and others** |
> | **MedVR (Ours)** | **OmniMedVQA (36K filtered)** | **RL** | **OmniMedVQA** |
>
> This table reveals a crucial point that we have now clarified in the paper: **the Lingshu-7B model was pre-trained and fine-tuned using data from both PMC-VQA and other medical datasets (12M)**. This means these benchmarks are not truly "out-of-domain" for Lingshu. In contrast, our MedVR framework was trained exclusively on a filtered subset of OmniMedVQA.
>
> **1.2 Context on Out-of-Domain Dataset Distributions**
>
> Furthermore, it is important to highlight the significant distributional differences between these datasets, which makes generalization inherently challenging.
>
> *   **OmniMedVQA** (our training set) is constructed from general **medical classification** datasets, covering a wide variety of modalities and VQA tasks.
>
> *   **PMC-VQA** is derived from image-caption pairs extracted from **online scientific papers**, creating a domain gap with images from real-world clinical applications.
>
> *   **MedXpertQA** is distinct as it incorporates rich **clinical information**, including patient records and examination results, moving beyond conventional images.
>
> Given these substantial distributional shifts, achieving strong OOD performance without direct training is a difficult task. Since Lingshu's massive pre-training corpus contains data similar to our training set and it was explicitly trained on the PMC-VQA training set, our model's ability to achieve directly comparable performance is a testament to its strong generalization capabilities. We apologize for the initial mischaracterization of "surpassing" and have revised the text to more accurately frame this compelling result.
>
> ---
>
> **2. New Controlled Experiments for Fair Comparison**
>
> To provide a more direct and unambiguous assessment of our method's contribution, we conducted a new set of **rigorously controlled experiments**. We applied our MedVR framework and a Textual RL baseline to a diverse set of backbone models (Qwen-3B/7B/32B and Lingshu-7B) using the **exact same** RL settings, rollout budget, and training corpora. This controlled setup isolates the impact of our proposed visual reasoning approach.
> | Model | Method | OmniMedVQA (In-domain) | PMC-VQA (OOD) | MedXpertQA (OOD) | Avg. |
> | --- | --- | --- | --- | --- | --- |
> | Qwen2.5-VL-3B | zero-shot | 63.9 | **50.6** | 21.9 | 45.5 |
> |  | Textual RL | **93.8** | 47.5 | 20.2 | 53.8 |
> |  | MedVR | 92.1 | 49.1 | **22.2** | **54.5** |
> | Qwen2.5-VL-7B | zero-shot | 64.3 | 51.2 | 22.3 | 46.0 |
> |  | Textual RL | 94.5 | 53.4 | 21.4 | 56.4 |
> |  | MedVR | **96.7** | **54.3** | **26.4** | **59.1** |
> | Qwen2.5-VL-32B | zero-shot | 69.9 | 54.1 | 24.5 | 49.5 |
> |  | Textual RL | 95.9 | 53.2 | 25.2 | 58.1 |
> |  | MedVR | **96.6** | **56.4** | **26.5** | **59.8** |
> | Lingshu-7B | zero-shot | 84.2 | 54.3 | 26.5 | 55.0 |
> |  | Textual RL | 95.7 | 53.7 | 25.1 | 58.3 |
> |  | MedVR | **96.0** | **60.8** | **26.5** | **61.1** |

---

> ### Author Response · Authors · 2025-11-27
> **Response to Reviewer 9mPg (2/3)**
>
> (Continuing from above) These results clearly demonstrate that **MedVR consistently and significantly outperforms the Textual RL baseline across all model sizes and backbones, with particularly strong gains on the OOD benchmarks**. This provides robust evidence for the universality and effectiveness of our visual reasoning framework. Notably, even on the already powerful Lingshu backbone, MedVR delivers a substantial boost in OOD performance, validating that the gains are attributable to our method itself, not confounding factors. We have added these results to **Section 4.5** in our revise manuscript.
>
> ---
>
> **3. New Evaluation on Free-Text VQA to Further Validate Generalization**
>
> To further demonstrate the generalizability of MedVR beyond multiple-choice formats, we conducted a new evaluation on three free-text VQA benchmarks. We used the train splits of VQA-RAD and SLAKE for training and evaluated on their test splits (in-domain) and the PathVQA test split (OOD).
> | Model | VQA-RAD | SLAKE | PathVQA (OOD) | Avg. |
> | --- | --- | --- | --- | --- |
> | Med-R1-2B | 39.0 | 54.5 | 15.3 | 36.3 |
> | MedVLM-R1-2B | 48.6 | 56.0 | 32.5 | 45.7 |
> | MedGemma-4B | 72.5 | 76.4 | 48.8 | 65.9 |
> | LLaVA-Med-7B | 53.7 | 48.0 | 38.8 | 46.8 |
> | HuatuoGPT-V-7B | 67.0 | 67.8 | 48.0 | 61.6 |
> | Lingshu-7B | 67.9 | 83.1 | 61.9 | 70.3 |
> | **MedVR (Ours)** | **74.4** | **85.3** | **62.3** | **74.0** |
>
> As shown, MedVR achieves new state-of-the-art results, significantly outperforming all other baselines on both in-domain and out-of-domain datasets. This provides further evidence of our method's effectiveness and robust generalization capabilities. We have added these new results and analyses to **Section 4.2** of the revised manuscript.
>
> ---
>
> **Response to W2 & Q1: Implementation Details of the zoom-in Tool**
>
> We apologize for the omission of these critical details and thank the reviewer for pointing it out. We have added a detailed explanation to **Appendix D.2**.
>
> 1.  **Input Image Resolution**: During training, to manage memory constraints, input images with a longest side exceeding 1024 pixels were resized to have a 1024-pixel longest side; smaller images retained their original resolution. During inference, we use the original, full resolution of the images.
>
> 2.  **Zoom-in Tool Workflow:** The `zoom-in` tool operates by cropping a region from the **original, high-resolution image**. This cropped region is then processed by the same pre-trained vision encoder used for the full image, generating a new set of visual tokens. These new tokens are bracketed by special markers (`<tool_response>` and `<\tool_response>`) and inserted into the model's context sequence. This ensures that the model's subsequent generation is conditioned on both its prior reasoning and this new, high-fidelity visual information. These observation tokens are masked out during the policy loss calculation as they are not generated by the policy.
>
> This clarification confirms that the tool performs a meaningful action by examining high-resolution image crops, not just low-resolution feature maps.
>
> ---
>
> **Response to W3 & Q2: Evaluation of Core Detection Capability**
>
> We agree that the reliability of the agent's visual grounding is fundamental to our method. To address this, we have conducted several new experiments.
>
> **1. Quantitative Evaluation of Localization Capability**
>
> We evaluated MedVR's localization performance on three distinct visual grounding tasks: General Medical VQA (GEMEX-ThinkVG [1]), Medical Phrase Grounding (ChestX-ray8 [2]), and Lesion Detection (ISIC [3]). We report the mean Intersection over Union (mIoU) between the predicted and ground-truth bounding boxes.
> | Model | GEMEX-ThinkVG | ChestX-ray8 | ISIC |
> | --- | --- | --- | --- |
> | Qwen2.5VL-7B (zero-shot) | 17.54 ± 2.13 | 36.53 ± 3.21 | 35.73 ± 1.87 |
> | **MedVR** | **59.62 ± 1.73** | **54.29 ± 1.81** | **69.12 ± 1.35** |
>
> The results show that the Qwen-7B backbone possesses some innate zero-shot localization ability and **our MedVR framework dramatically improves this capability**. This demonstrates that our annotation-free training process effectively teaches the model to accurately localize clinically relevant regions, providing a reliable foundation for its reasoning.

---

> ### Author Response · Authors · 2025-11-27
> **Response to Reviewer 9mPg (3/3)**
>
> **2. Ensuring Zoom-in Actions Target Clinically Relevant Regions**
>
> "How we can be sure that `zoom-in` actions are targeting clinically relevant regions" is precisely the challenge our method is designed to solve. A naive agent does not inherently know where to look. As shown in our ablation study  (**Table 2**), simply providing the `zoom-in` tool without our EVR and CCA mechanisms leads to suboptimal performance. We designed our framework to address this:
>
> *   **Entropy-guided Visual Regrounding (EVR)**: To empirically validate EVR's hypothesis that token-level entropy is a reliable proxy for visual grounding uncertainty, we conducted a new analysis on the GEMEX-ThinkVG dataset. We measured the entropy of tool-call tokens from the Qwen2.5-VL-7B and the corresponding mIoU of the generated box. As shown in **Figure 3** of our revised manuscript, tool calls resulting in high-quality localizations (high mIoU) exhibit significantly lower entropy. This confirms that EVR effectively guides the model to explore when it is uncertain, improving localization quality.
>
> *   **Consensus-based Credit Assignment (CCA):** To validate that our annotation-free CCA mechanism finds a meaningful "consensus" and does not simply amplify noise, we compared it against a supervised upper bound. On the GEMEX-ThinkVG dataset, we ran an experiment where the CCA consensus mask was replaced with the ground-truth mask, providing direct supervision for the `zoom-in`  location. The performance of our annotation-free approach is remarkably close to that of its fully supervised counterpart. This provides strong evidence that CCA is highly effective at distilling a high-quality reward signal from rollouts and guiding the model toward semantically relevant regions, thus preventing the "poisoning" of the reward signal.
> | Method | Accuracy (%) | mIoU (%) |
> | --- | --- | --- |
> | MedVR (w/ Annotation) | 79.62 | 61.33 |
> | MedVR (w/o Annotation) | 79.08 | 59.62 |
>
> ---
>
> **Response to W4 & Q3: Model Scaling Analysis**
>
> We thank the reviewer for this suggestion. Our new controlled experiments, presented in the table under the first point (**Response to W1, W5, Q4: Misrepresentation, OOD Performance, and Fairness of Comparisons**), directly address the question of model scaling. We evaluated MedVR on Qwen2.5VL backbones of 3B, 7B, and 32B parameters. The results lead to two clear conclusions:
>
> 1.  **Consistent Improvement Across Scales**: MedVR consistently achieves higher average scores than the Textual RL baseline across all model sizes (3B, 7B, and 32B), demonstrating that our visual reasoning framework is a general approach that is not finicky or specific to one architecture.
>
> 2.  **Enhanced OOD Generalization**: The performance gains from MedVR are particularly pronounced on OOD benchmarks. In stark contrast, the Textual RL baseline exhibits a significant performance degradation on these OOD tasks, indicating a tendency to overfit to the source domain. This finding further reinforces the critical role of grounded visual reasoning in achieving robust model generalization.
>
> This scaling analysis confirms that MedVR is a robust and scalable framework that enhances the model's ability to generalize its reasoning skills, a critical capability for real-world medical applications.
>
> ---
>
> **Thank you for your valuable comment again, we hope our responses can address your concern and we look forward to your further reply!**
>
> **Reference:**
>
> [1] GEMeX-RMCoT: An Enhanced Med-VQA Dataset for Region-Aware Multimodal Chain-of-Thought Reasoning. ACMMM, 2025.
>
> [2] ChestX-ray8: Hospital-Scale Chest X-Ray Database and Benchmarks on Weakly-Supervised Classification and Localization of Common Thorax Diseases. CVPR, 2017.
>
> [3] Skin lesion analysis toward melanoma detection 2018: A challenge hosted by the international skin imaging collaboration (isic). arXiv, 2019.

---

### Official Review · Reviewer_P4Jp · 2025-10-29

**Soundness:** 4
**Presentation:** 3
**Contribution:** 4
**Rating:** 8
**Confidence:** 3

**Summary:**

This paper introduces an innovative approach for training medical vision-language models through an annotation-free reinforcement learning framework. The authors demonstrate that reinforcement learning can effectively supervise model behavior using only final ground-truth answers, while incorporating a consensus-driven self-supervision mechanism that leverages model convergence across multiple reasoning trajectories. This design enables the model to learn fine-grained visual reasoning without relying on costly manual annotations. I find the work highly original and thoughtfully executed—the proposed combination of uncertainty-guided exploration and consensus-based reward shaping represents a creative step forward in medical AI. Overall, this is a well-motivated, technically solid, and enjoyable paper to read.

**Strengths:**

- The paper is clearly written and easy to follow, with strong organization and clear motivation.

- It presents a technically rigorous method, combining reinforcement learning, uncertainty-based exploration, and self-supervised reward shaping in a coherent framework.

- The approach offers a novel way to guide RL toward fine-grained visual regions, ensuring that the model’s reasoning is grounded in meaningful image details rather than text-only patterns.

- Empirical results are strong and consistent across multiple benchmarks

**Weaknesses:**

- Reproducibility: Please release the code repository. For a complex RL-based framework like this, access to the full codebase and training scripts is essential for replication and community adoption.

- Evaluation scope: The paper primarily evaluates on multiple-choice medical VQA datasets, which simplifies the scoring through accuracy. How would the proposed RL and credit-assignment framework adapt to open-ended or free-text medical reasoning questions where correctness is less binary?

- Training initialization:
a) The paper claims annotation-free RL training, yet the model produces structured reasoning traces (<think>, <tool_call>, <answer>). Was a supervised fine-tuning (SFT) or instruction-tuning stage used to teach this output format before RL?
b) If so, could the authors clarify what data or synthetic prompts were used for this warm-up, and whether this step contributes to the strong structured reasoning behaviors observed in Figure 3?

**Questions:**

See weakness

---

> ### Author Response · Authors · 2025-11-27
> **Response to Reviewer P4Jp**
>
> We sincerely thank the reviewer for your insightful feedback and constructive suggestions, which have helped us to improve the quality of our manuscript. We are encouraged that the reviewer found our work promising. Below, we provide a point-by-point response to the weaknesses and questions raised.
>
> ---
>
> **Response to W1: Reproducibility**
>
> We thank the reviewer for emphasizing the importance of reproducibility, which we agree is a cornerstone of scientific research. We are fully committed to ensuring that our work is accessible and verifiable by the research community. We will **release the complete codebase, pre-trained models, and data processing scripts upon paper acceptance**. Additionally, we will provide detailed documentation of our experimental settings and hyperparameter configurations to facilitate a deeper understanding and application of our work.
>
> ---
>
> **Response to W2: Evaluation Scope**
>
> We thank the reviewer for this excellent question regarding the generalizability of our framework to open-ended tasks. While multiple-choice VQA allows for straightforward accuracy-based evaluation, we agree that demonstrating efficacy on free-text generation tasks is also crucial for validating the broader applicability of our RL and credit-assignment mechanisms.
>
> To address this, we have conducted new experiments evaluating MedVR on free-text medical VQA task. We utilized the training splits of VQA-RAD and SLAKE for training. Their respective test splits were used for in-domain evaluation, while the PathVQA dataset was used to assess the out-of-domain generalization capability. To adapt MedVR to open-ended generation, we reformulated the accuracy reward `R_acc`. Instead of binary correctness, we designed a composite reward function based on a combination of widely-used text generation metrics: BLEU, ROUGE, and BERTScore following prior work [1]. As shown in the table below, MedVR significantly outperforms all baseline models across both in-domain and out-of-domain settings.
> | Model | VQA-RAD | SLAKE | PathVQA | Avg. |
> | --- | --- | --- | --- | --- |
> | Med-R1-2B | 39.0 | 54.5 | 15.3 | 36.3 |
> | MedVLM-R1-2B | 48.6 | 56.0 | 32.5 | 45.7 |
> | MedGemma-4B | 72.5 | 76.4 | 48.8 | 65.9 |
> | LLaVA-Med-7B | 53.7 | 48.0 | 38.8 | 46.8 |
> | HuatuoGPT-V-7B | 67.0 | 67.8 | 48.0 | 61.6 |
> | Qwen2.5VL-7B | 64.5 | 67.2 | 44.1 | 58.6 |
> | InternVL2.5-8B | 59.4 | 69.0 | 42.1 | 56.8 |
> | InternVL3-8B | 65.4 | 72.8 | 48.6 | 62.3 |
> | Lingshu-7B | 67.9 | 83.1 | 61.9 | 70.3 |
> | MedVR (Ours) | **74.4** | **85.3** | **62.3** | **74.0** |
>
> These results provide strong evidence that our framework can be effectively adapted to the more challenging domain of open-ended VQA scenario, where it successfully learns to generate accurate and relevant free-text responses by grounding its reasoning in visual evidence. We have added these results to **Section 4.2**.
>
> ---
>
> **Response to W3: Training Initialization**
>
> We appreciate the reviewer's request for clarification on our training methodology and the "annotation-free" claim. We have provided further clarification on this in **Appendix D.1**.
>
> 1.  We would like to confirm that our approach **does not use any supervised fine-tuning (SFT) or instruction-tuning on human-annotated reasoning traces to teach the output format**. Our claim of "annotation-free RL training" refers to the fact that the intermediate reasoning steps (i.e., the visual grounding boxes) are learned without explicit ground-truth annotations. The model learns to produce structured outputs directly through reinforcement learning.
>
> 2.  The structured reasoning format is achieved by providing a detailed **system prompt** to the base model, which explicitly defines the expected structure for thoughts and tool calls, similar to methodologies used in agentic AI frameworks. We instruct the model to call the `image_zoom_in_tool` using a specific JSON format:
> `{"name": "image_zoom_in_tool", "arguments": {"bbox_2d": [x1, y1, x2, y2], "label": " nodules"}}`.
> The strong instruction-following capabilities of the Qwen2.5-VL-7B base model allow it to understand and adhere to this predefined format during the RL process without requiring a separate SFT warm-up stage. Therefore, the structured reasoning behaviors observed in Figure 3 are a direct result of the RL agent learning to operate within the constraints defined by the system prompt, guided by the task and tool rewards, rather than from any pre-training on structured data.
>
> ---
>
> **Thank you for your valuable comment again, we hope our responses can address your concern and we look forward to your further reply!**
>
> **Reference:**
>
> \[1\] Improving Medical Reasoning with Curriculum-Aware Reinforcement Learning. Arxiv, 2025.

---

### Official Review · Reviewer_dA77 · 2025-11-07

**Soundness:** 3
**Presentation:** 2
**Contribution:** 2
**Rating:** 4
**Confidence:** 3

**Summary:**

This paper presents MedVR, a reinforcement learning framework that enables annotation-free visual reasoning in medical vision-language models. It introduces two mechanisms: Entropy-guided Visual Regrounding (EVR), which uses model uncertainty to trigger targeted visual exploration, and Consensus-based Credit Assignment (CCA), which derives pseudo-supervision from agreement among successful reasoning trajectories. Without human annotations, MedVR achieves state-of-the-art performance on multiple medical VQA benchmarks, improving both accuracy and visual grounding in clinical reasoning tasks.

**Strengths:**

- Proposes an annotation-free visual reasoning framework for medical VLMs
- The idea of zooming into medical images to guide reasoning is inspired by real clinical practice, making the approach both creative and intuitively grounded
- Using entropy as a proxy for uncertainty to trigger visual exploration adds interpretability to the reasoning process
- Demonstrates consistent state-of-the-art performance across multiple medical VQA benchmarks, supported by detailed ablations validating each component
- Addresses a key limitation in medical AI—lack of fine-grained visual supervision—in a practical and scalable way

**Weaknesses:**

- The use of entropy as a trigger for zoom-in is interesting but under-justified. The paper could provide stronger evidence that high-entropy points truly correspond to moments where zooming in improves reasoning. Visualizations or quantitative correlations between entropy spikes and successful zoom-ins would make this claim more convincing.

- The method appears particularly well-suited for localization, lesion detection, or report generation, where visual grounding and fine-grained inspection matter most. However, these tasks are not evaluated, limiting the evidence that the approach generalizes beyond multiple-choice VQA.

- It seems that every entropy spike results in a zoom-in, but the paper does not show when or how often this happens. It would be helpful to analyze where in the reasoning chain these entropy peaks occur and whether those zooms actually lead to better answers.

- Tool-based visual reasoning is becoming increasingly common (e.g., Chain-of-Focus, Pixel Reasoner, DeepEyes). This paper mainly introduces the use of a zoom-in tool for the medical domain rather than a generally novel method. Therefore, it can feel more like a heuristic engineering improvement than a conceptual breakthrough. While combining entropy-based exploration and consensus pseudo-labeling is clever, both ideas are extensions of known techniques rather than fundamentally new principles.

To make the contribution stronger, the authors could either (1) demonstrate that the approach works on broader visual reasoning tasks including on general domains, or (2) position it as a domain-specific contribution for a medical AI venue.

Overall: The idea is interesting and well-motivated, especially for medical settings, but needs stronger justification, broader evaluation, and clearer evidence that entropy-driven zooming genuinely improves reasoning.

**Questions:**

- Can you provide evidence that entropy spikes truly indicate regions where zooming in is useful? For example, visualize entropy over time or correlate entropy with answer improvements.

- How frequently and at what stages do entropy spikes occur during reasoning? A simple distribution or timeline would clarify how often zoom-ins are triggered.

- The approach seems well-suited for localization, detection, or report generation—why not evaluate on those tasks?

- Would EVR + CCA generalize to non-medical visual reasoning tasks, or is it intended primarily for medical AI applications?

---

> ### Author Response · Authors · 2025-11-27
> **Response to Reviewer dA77 (1/3)**
>
> We sincerely thank the reviewer for your thoughtful and constructive feedback. Your comments have helped us identify key areas for clarification and improvement. We are particularly grateful for the opportunity to provide a more robust justification for our proposed method and to demonstrate its broader applicability.
>
> **Response to W1, W3, Q1, Q2: Justification and Mechanics of the Entropy-Guided Visual Regrounding (EVR)**
>
> We appreciate the reviewer's insightful questions regarding the role and efficacy of entropy in our framework. We will address these concerns together, as they are closely related.
>
> **1. Clarification on the Role of Entropy in EVR**
>
> First, we would like to clarify a potential misunderstanding regarding the role of entropy. In MedVR, **entropy is not a direct trigger for the** `zoom-in` operation itself. Rather, the decision to use the `zoom-in` is **learned autonomously** by the agent based on the input query and image.
>
> The primary motivation for designing EVR was to address a core challenge in agent-based visual reasoning: _"Without explicit supervision, an agent may struggle to learn where to look, often resorting to random or repetitive visual actions that are computationally expensive and yield little information."_ As our ablation study in **Table 2** demonstrates, simply providing the `zoom-in` tool without our proposed learning mechanisms leads to suboptimal performance.
>
> Our goal was to encourage efficient and meaningful **exploration**. We hypothesized that the model's confidence in its visual actions (i.e., where to zoom) could serve as a powerful signal. We use entropy as a quantitative measure of the model's uncertainty regarding its visual grounding decisions.
>
> *   When the model is **confident** (low entropy) about a zoom-in location, we preserve this high-quality action.
>
> *   When the model is **uncertain** (high entropy), EVR triggers **adaptive branching** to encourage exploration of alternative visual hypotheses.
>
> In summary, entropy is not the trigger for **whether to zoom** — the decision  is made autonomously by the model — but rather the trigger for **initiating a new round of reasoning to explore alternative zoom locations** when the model is uncertain about its visual decisions.
>
> **2. Evidence of Correlation between Entropy and Localization Quality**
>
> To provide direct evidence that entropy spikes indicate moments where refined exploration is beneficial, we have conducted a new analysis and included it as  **Figure 3** in the revised manuscript. We analyzed all tool-call tokens generated by the Qwen2.5-VL-7B on the GEMEX-ThinkVG [1] dataset, which provides grounding box annotations for visual references. We computed the average generation entropy of these tokens across different intervals of Intersection over Union (IoU) between the predicted and ground-truth bounding boxes.
>
> As shown, tool calls resulting in high-quality localizations (higher IoU) exhibit significantly lower entropy. Conversely, low-IoU regions are associated with high-entropy generation steps. This empirically validates our core hypothesis:  **entropy serves as a reliable proxy for the model's uncertainty in its visual grounding decisions**. This evidence justifies our use of high-entropy moments to trigger adaptive branching, as it directs exploration precisely when the model's localization is likely to be suboptimal.
>
> **3. Entropy Dynamics and Training Stability**
>
> To illustrate how EVR influences the training process, we have added **Figure 4** to the paper. This figure tracks the average token entropy during training for models trained with and without the EVR mechanism.
>
> We observe that EVR **maintains consistently higher and more stable entropy levels throughout training**. This prevents the model from entropy collapse, a phenomenon where the policy prematurely converges to a suboptimal, low-exploration state. This sustained exploration is crucial for discovering more robust and generalizable reasoning strategies, which in turn facilitates steady and reliable performance improvements.
>
> **4. Frequency of Entropy-Driven Exploration**
>
> The frequency of entropy-driven branching is co-determined by the exploration budget (the number of trajectories reserved for branching) and the entropy weight \gamma. In our default experimental setup,  we observed that nearly 50% of the trajectories were triggered by entropy spikes. This indicates that the mechanism is frequently engaged to resolve model uncertainty.

---

> ### Author Response · Authors · 2025-11-27
> **Response to Reviewer dA77 (2/3)**
>
> **5. Summary of EVR Advantages**
>
> To conclude, the advantages of our EVR are threefold:
>
> *   **Enhanced Performance:** By dynamically adapting exploration based on uncertainty, EVR guides the model to gather more informative visual evidence. This leads to improved localization quality and overall reasoning performance, which is particularly evident in the superior OOD results shown in Table 2 of our paper.
>
> *   **Training Stability:** As demonstrated in our new Figure 8, EVR prevents premature convergence by maintaining healthy entropy levels, ensuring the model continues to explore diverse visual strategies.
>
> *   **Computational Efficiency:** EVR's tree-search exploration is more efficient than naively generating multiple independent trajectories. By sharing prefixes across branches, it eliminates redundant token generation and reduces tool call overhead. This adaptive branching approach effectively reuses already-generated tokens at branching points rather than regenerating entire sequences, significantly reducing the overhead associated with tool calls. Consequently, the computational complexity is potentially reduced from $ O(n^2) $ to $ O(n \log n) $. As quantitatively evaluated in our new **Figure 5**, EVR substantially lowers computational costs.
>
> We gratefully acknowledge the reviewer for identifying key aspects of our method that contribute to enhancing the quality of this paper.
>
> ---
>
> **Response to W3, Q2: Distribution of Tool Use**
>
> To clarify how often the `zoom-in` tool is used, we have added **Figure 8** to **Appendix A.6**, which shows the average number of tool calls per trajectory during the training process. The trend shows that the average number of tool calls initially increases as the model learns to leverage the tool, then decreases and stabilizes at approximately 1.05 calls per trajectory. This demonstrates that the model does not randomly call the tool. Instead, through reinforcement learning, it **learns to use** the `zoom-in` function more judiciously and effectively as training progresses, mastering when and how to apply it to improve its reasoning.
>
> ---
>
> **Response to W2, Q3: Evaluation on Tasks Beyond Multiple-Choice VQA**
>
> We agree with the reviewer that demonstrating generalization beyond multiple-choice VQA is crucial for substantiating our claims. To address this, we have conducted extensive new experiments on three additional tasks: free-text VQA, medical phrase grounding, and lesion detection. MedVR achieves significant improvements over strong baselines across all tasks, showcasing its versatility and robustness. We have included these results in the appendix of the revised manuscript. Please check the updated version for details.
>
> **1. Free-Text VQA**
>
> We trained on the VQA-RAD [2] and SLAKE [3] training splits and evaluated on their respective test sets (in-domain) and on PathVQA [4] (out-of-domain). Following prior work[5], we used a composite reward of BLEU-1, ROUGE-1, and BERTScore, as illustrated in **Appendix D.3**.
> | Model | VQA-RAD | SLAKE | PathVQA | Avg. |
> | --- | --- | --- | --- | --- |
> | Med-R1-2B | 39.0 | 54.5 | 15.3 | 36.3 |
> | MedVLM-R1-2B | 48.6 | 56.0 | 32.5 | 45.7 |
> | MedGemma-4B | 72.5 | 76.4 | 48.8 | 65.9 |
> | LLaVA-Med-7B | 53.7 | 48.0 | 38.8 | 46.8 |
> | HuatuoGPT-V-7B | 67.0 | 67.8 | 48.0 | 61.6 |
> | Qwen2.5VL-7B | 64.5 | 67.2 | 44.1 | 58.6 |
> | InternVL2.5-8B | 59.4 | 69.0 | 42.1 | 56.8 |
> | InternVL3-8B | 65.4 | 72.8 | 48.6 | 62.3 |
> | Lingshu-7B | 67.9 | 83.1 | 61.9 | 70.3 |
> | **MedVR (Ours)** | **74.4** | **85.3** | **62.3** | **74.0** |
>
> **2. Medical Phrase Grounding**
>
> We evaluated on the ChestX-ray8 [6] dataset. MedVR demonstrates superior ability to localize regions from textual descriptions.
> | Model | mIoU | Acc |
> | --- | --- | --- |
> | Lvit | 33.50 | 35.05 |
> | GuideDecoder | 32.70 | 34.68 |
> | MedRPG | 34.59 | 38.02 |
> | RecLMIS | 33.05 | 40.46 |
> | ChEX | 38.49 | 41.60 |
> | CausalCLIPSeg | 40.12 | 46.02 |
> | Qwen2.5-VL-7B | 36.53 | 39.34 |
> | **MedVR (Ours)** | **54.29** | **64.36** |
>
> **3. Lesion Detection**
>
> We evaluated our method on the ISIC [7] dataset for skin lesion detection, and MedVR still achieves strong performance.
> | Model | mIoU | Acc |
> | --- | --- | --- |
> | Qwen2.5-VL-7B | 35.73 | 38.49 |
> | **MedVR (Ours)** | **69.12** | **72.21** |
>
> Regarding **report generation**, we acknowledge its importance. However, this task presents unique challenges in a zero-RL setting, particularly in designing verifiable rewards that capture clinical accuracy and narrative coherence. As zero-RL baselines for this specific task are not yet established in the medical domain, we have deferred its evaluation. We plan to explore effective reward designs for report generation in future work.

---

> ### Author Response · Authors · 2025-11-27
> **Response to Reviewer dA77 (3/3)**
>
> **Response to  W4, Q4: Generalization to Non-Medical Domains and Novelty of Our Method**
>
> **1. Generalization of EVR and CCA to Non-Medical Tasks**
>
> Thank you for your suggestion. To prove that our method is a general-purpose improvement, we evaluated our core mechanisms (EVR and CCA) on general-domain benchmarks. We trained on RefCOCO/+/g [8] and Geometry3k [9] and tested on the grounding and multimodal reasoning benchmarks respectively.
>
> **Grounding Benchmarks**
>
> | Model | refCOCO | refCOCO+ | refCOCOg | Avg. |
> | -- | -- | -- | -- | -- |
> | Qwen2.5-VL-7B | 89.1 | 82.6 | 86.1 | 85.6 |
> | DeepEyes | 89.8 | 83.6 | 86.7 | 86.7 |
> | **Ours** | **92.1** | **85.2** | **87.4** | **88.2** |
>
> **Multimodal Reasoning Benchmarks**
>
> | Model | MathVision | MathVerse | MathVista | Avg. |
> | -- | -- | -- | -- | -- |
> | Qwen2.5-VL-7B | 24.9 | 43.8 | 66.3 | 44.3 |
> | DeepEyes | 26.6 | **47.3** | 70.1 | 47.9 |
> | **Ours** | **29.0** | 46.4 | **71.4** | **48.9** |
>
> The results confirm that EVR and CCA are indeed generalizable techniques that enhance visual reasoning on non-medical tasks, outperforming both the strong backbone model and specialized visual reasoning models like DeepEyes[10].
>
> ---
>
> **2. Clarification of Our Method's Novelty and Contribution**
>
> We appreciate the reviewer's perspective on the broader field of tool-based visual reasoning and have cited these relevant works in **Appendix C**. However, we wish to respectfully clarify the primary conceptual novelty of our work, which we believe is more than a "heuristic engineering improvement".
>
> The core contribution of MedVR is its exploration of how to achieve  **meaningful visual grounding in an annotation-free setting**. Prior works in general domains, such as Pixel Reasoner [11], ViGoRL [12], and VGR [13], critically depend on **large-scale, human-annotated SFT datasets** with fine-grained grounding labels to bootstrap the model's tool-use capabilities. This requirement is prohibitively expensive and often infeasible in the medical domain, where such annotations require specialized expertise and are scarce.
>
> Our work is the first to tackle the challenge of learning effective visual grounding for medical VQA **without any human-annotated grounding data**. MedVR leverages only verifiable end-task rewards and the model's own internal signals through our proposed EVR (for uncertainty-driven exploration) and CCA (for consensus-based pseudo-labeling) mechanisms. This shift from reliance on external, fine-grained annotations to self-generated supervision for intermediate reasoning steps represents a significant conceptual advance.
>
> Therefore, we believe MedVR offers both: it is **(1) a domain-specific contribution** that provides a principled and practical solution to the critical annotation scarcity problem in medical AI, and this solution is built upon **(2) a generally applicable approach** to annotation-free visual reasoning, as evidenced by our new experiments.
>
> ---
>
> In summary, we have provided substantial new evidence and extensive experiments on new tasks and domains, to address the weaknesses and questions raised. We have clarified that entropy guides exploration, and provided empirical data to justify this design. We have also demonstrated our method's versatility beyond multiple-choice VQA and its novelty in enabling annotation-free visual reasoning. We believe these additions significantly strengthen our paper.
>
> **We thank you once again for your valuable and constructive feedback. We hope our responses can address your concern and we look forward to your further reply!**
>
> **Reference:**
>
> [1] GEMeX-RMCoT: An Enhanced Med-VQA Dataset for Region-Aware Multimodal Chain-of-Thought Reasoning. ACMMM, 2025.
>
> [2] A dataset of clinically generated visual questions and answers about radiology images. Scientific Data, 2018.
>
> [3] Slake: A semantically-labeled knowledge-enhanced dataset for medical visual question answering. ISBI, 2021.
>
> [4] Pathvqa: 30000+ questions for medical visual question answering. Arxiv, 2020.
>
> [5] Improving Medical Reasoning with Curriculum-Aware Reinforcement Learning. Arxiv, 2025.
>
> [6] ChestX-ray8: Hospital-Scale Chest X-Ray Database and Benchmarks on Weakly-Supervised Classification and Localization of Common Thorax Diseases. CVPR, 2017.
>
> [7] Skin lesion analysis toward melanoma detection 2018: A challenge hosted by the international skin imaging collaboration (isic). arXiv, 2019.
>
> [8] Generation and comprehension of unambiguous object descriptions. ICCV, 2016.
>
> [9] Inter-gps: Interpretable geometry problem solving with formal language and symbolic reasoning. Arxiv, 2021.
>
> [10] DeepEyes: Incentivizing “Thinking with Images” via Reinforcement Learning. Arxiv, 2025.
>
> [11] Pixel Reasoner: Incentivizing Pixel-Space Reasoning with Curiosity-Driven Reinforcement Learning. NeurIPS, 2025.
>
> [12] UniVG-R1: Reasoning Guided Universal Visual Grounding with Reinforcement Learning. arXiv, 2025.
>
> [13] VGR: Visual Grounded Reasoning. arXiv, 2025.

---

### Official Review · Reviewer_iJaK · 2025-11-07

**Soundness:** 3
**Presentation:** 3
**Contribution:** 3
**Rating:** 6
**Confidence:** 3

**Summary:**

This work aims to propose a framework to better ground the reasoning of vision-language models into the visual perception. Concretely, 2 modules are proposed: (1) a entropy-guided visual grounding module using summed token entropy based model uncertainty (2) a consensus-based credit assignment module using distilled pseudo supervision from rollout agreement (e.g., union of multiple proposed bounding boxes). A qualitative study is demonstrated and quantitative experimental results show superior performance in medical VQA tasks without using human annotations.

**Strengths:**

- The writing is easy to follow.
- The designed uncertainty based spatial exploration strategy based on the sum of entropy is interesting.
- The consensus module is carefully designed as a posterior refinement step.

**Weaknesses:**

- The usage of “evidence” in this paper is not consistent with what “evidence” means in clinical practice. In clinical setting, “evidence” means the best available external clinical evidence from systematic research, instead of from the current tested image [1].

[1] Sackett D L. Evidence-based medicine[C]//Seminars in perinatology. WB Saunders, 1997, 21(1): 3-5.

-	The contribution on “grounding every inferential step in verifiable visual evidence” is overclaimed. Is “zoom-in” enough to ground “every inferential step”? Is the “zoom-in” area always correct? What does “verifiable visual evidence” mean compared to “visual evidence”?
-	The image manipulation tool of the approach is limited only to “zoom-in”. A wide range of other image manipulation tools such as different filtering/denoising techniques are not explored.
-	There is no experiment justifying whether the zoom-in area is actually meaningful or not.
-	It’s unclear whether this annotation-free approach might perform better than using the annotation or not under a fair comparison’s setting.

-	[line 104] The claim “First to endow medical VLMs with explicit visual reasoning capabilities.” is somewhat vague. This approach offers some improvements in the VQA performance, but existing powerful large MLLM such as GPT4o can also be deployed for medical tasks and has visual reasoning capabilities. What “explicit” means is unclear neither.

**Questions:**

[line 107] what clinical workflow is referenced here?

[line 162-164] Some implementation details are not clear enough to me: After a cropped image is obtained, is the image up-sampled to recover the full resolution or do they stay with their original resolution? How are the visual tokens of this cropped area obtained? Any additional padding? How are these visual tokens inserted into the reasoning path? What’s the latency of doing this in inference time?

Will this agent crop images multiple times at different scales in different areas in a “successful trajectory”? If yes, would these areas contradict with each other (e.g., no overlapping, indicating the model is looking at completely different areas)?

---

> ### Author Response · Authors · 2025-11-27
> **Response to Reviewer iJaK (1/3)**
>
> We are very grateful to the reviewer for your thorough and insightful feedback. The questions have helped us to clarify important aspects of our work and strengthen the manuscript. We have addressed each point in detail below.
>
> **W1: On the Usage of "Evidence"**
>
> We thank the reviewer for this important and precise clarification regarding the term "evidence". We agree that our use of "visual evidence" could be misinterpreted in the context of Evidence-Based Medicine (EBM).
>
> Our intention was to use this term in a manner consistent with the computer vision and AI literature, where it refers to the **perceptual data within an image that supports a model's inference**. Our goal is not to redefine the established meaning of "clinical evidence" but to make the model's reasoning process more transparent by compelling it to ground its textual inferences in specific, observable features of the input image. This process is analogous to the visual inspection phase of a clinician's diagnostic workflow, where they identify findings on an image to support a hypothesis.
>
> To prevent any misunderstanding and better align with clinical terminology, we have revised the manuscript to use more precise phrasing **[Line 46-48]**. This will more accurately reflect our contribution, which is focused on enhancing the model's interpretability by linking its outputs to the source image, rather than making claims about clinical evidence in the EBM sense.
>
> **W2: On the Overclaimed Contribution of "Grounding Every Inferential Step in Verifiable Visual Evidence"**
>
> We appreciate the reviewer's request for clarification on this central claim. We acknowledge that the word "every" is an overstatement and have revised it to more accurately reflect our method's behavior **[Line 46-48]**.
>
> **W2.1: "Is 'zoom-in' enough for 'every inferential step'?"**
>
> Our model interleaves textual deliberation with visual tool use. The claim is not that every single generated token is grounded by a zoom action, but that the **critical visual analysis steps** within the reasoning chain are explicitly grounded. In practice, the model still leverages its extensive medical knowledge through textual deliberation. The `zoom-in` tool is invoked selectively when the model determines that a closer look is needed to verify a hypothesis or refine its visual analysis. We believe that providing verifiable visual grounding for these key decision points significantly enhances the trustworthiness and interpretability of the model's output.
>
> **W2.2: "Is the 'zoom-in' area always correct?"**
>
> This is precisely the challenge our method is designed to address. A naive agent does not inherently know where to look. As our ablation study in **Table 2** demonstrates, simply providing the `zoom-in` tool without our proposed learning mechanisms leads to suboptimal performance.
>
> To solve this, we designed Entropy-guided Visual Regrounding (EVR) and Consensus-based Credit Assignment (CCA) to optimize the selection of zoom-in areas during RL training, ensuring the model learns to identify and focus on meaningful and causally relevant regions. The qualitative examples in **Figures 6 and 7** of our revised paper show that these learned zoom-in areas correspond to clinically significant findings.
>
> To provide further quantitative validation, we conducted a new experiment to evaluate the localization quality on three different tasks: General Medical VQA (GEMEX-ThinkVG [1] ), Medical Phrase Grounding (ChestX-ray8 [2] ), and Lesion Detection (ISIC [3] ). We report the mean Intersection over Union (mIoU) between the localized regions and the ground truth.
> | Model | GEMEX-ThinkVG | ChestX-ray8 | ISIC |
> | --- | --- | --- | --- |
> | Qwen2.5-VL-7B (zero-shot) | 17.54 ± 2.13 | 36.53 ± 3.21 | 35.73 ± 1.87 |
> | **MedVR** | **59.62 ± 1.73** | **54.29 ± 1.81** | **69.12 ± 1.35** |
>
> These results demonstrate that MedVR learns to localize relevant regions with significant accuracy. We add these results to **Appendix  A.2** to further showcase the quality of the generated zoom-in areas.
>
> **W2.3: "What does 'verifiable visual evidence' mean?"**
>
> For a standard medical VLM, the visual basis for a textual conclusion is typically latent, hidden within its internal attention mechanisms. In contrast, MedVR externalizes this grounding step by generating an explicit bounding box and a cropped image patch as part of its reasoning trace. This allows a human expert to look at the exact region the model examined and "verify" whether the model's subsequent reasoning is plausible based on that specific visual input. This explicit, auditable link between the model's reasoning and observable image features is what we mean by "verifiable".

---

> ### Author Response · Authors · 2025-11-27
> **Response to Reviewer iJaK (2/3)**
>
> **W3: Limited Image Manipulation Tools**
>
> We thank the reviewer for this point. We agree that exploring a broader set of tools is a valuable direction for future work, but our choice of the `zoom-in` tool was deliberate and strategic for the following reasons:
>
> 1.  **Focus on Annotation-Free Learning**: Our primary research contribution is the annotation-free framework for visual reasoning, which is orthogonal to the specific tools used. Our agentic RL framework is flexible and can be extended to a more diverse toolset without altering its core principles.
>
> 2.  **Strategic Rationale for** `zoom-in`:
>
>     *   **Leveraging Internal Capabilities**: Unlike tools such as filtering or denoising, which are external image processing functions, `zoom-in` is designed to leverage and enhance the model's **internal grounding capabilities**. Pre-trained VLMs already possess some ability to localize objects. Our goal is to guide and reinforce this innate ability for visual reasoning, rather than simply teaching the model to call external APIs. The significant improvement in localization quality (W2.2) shows that our method successfully enhances the model's own reasoning skills.
>
>     *   **Alignment with Clinical Workflow**: The `zoom-in` action closely mimics a fundamental step in a clinician's diagnostic process, where they magnify specific regions of an image to inspect details. This makes the tool naturally applicable and interpretable in the medical domain.
>
> **W4: Whether the zoom-in Area Is Actually Meaningful**
>
> We respectfully refer the reviewer to our response to **W2.2** above, which provides a detailed discussion and new quantitative results demonstrating the meaningfulness of the zoom-in areas selected by our model.
>
> **W5: Comparison with an Annotation-based Method**
>
> To investigate this, we conducted a new experiment on the GEMEX-ThinkVG dataset, which provides ground-truth masks. We compared our annotation-free MedVR against a version where the CCA consensus mask was replaced with the ground-truth mask, providing direct supervision for the `zoom-in` location.
> | Method | Accuracy (%) | mIoU (%) |
> | --- | --- | --- |
> | MedVR (w/ Annotation) | 79.62 | 61.33 |
> | MedVR (w/o Annotation) | 79.08 | 59.62 |
>
> Notably, while using ground-truth annotations yields a modest improvement in localization quality (mIoU), the performance of our annotation-free approach is highly comparable to that of its fully supervised counterpart. It proves that our method is highly effective at guiding the model to semantically relevant regions. We have added these results in **Appendix A.3** of our revised manuscript.
>
> Most importantly, the core motivation for our work stems from the reality that fine-grained grounding annotations are prohibitively expensive and largely unavailable for the vast majority of medical imaging data. A large-scale, multi-modal, multi-disease annotated dataset for visual reasoning does not yet exist. Therefore, our primary contribution is not to outperform a hypothetical fully-supervised model, but to provide a scalable and practical solution that achieves robust visual reasoning **without requiring costly annotations**.
>
> **W6: The Claim of Being "First to Endow Medical VLMs with Explicit Visual Reasoning Capabilities"**
>
> We thank the reviewer for pointing out this ambiguity. We will clarify this claim by providing a more precise definition of "explicit visual reasoning" in our context.
>
> By "explicit," we refer to a reasoning process where the model **actively generates executable tool commands** to manipulate its visual input and then **integrates the output of these actions back into its reasoning chain**. This creates an observable, step-by-step trajectory of interleaved thought and action.
>
> While powerful general-purpose models like GPT-4o possess remarkable visual understanding, their reasoning is typically an implicit, monolithic process that generates a textual chain-of-thought based on a static image. They do not perform this agentic, tool-augmented reasoning loop. To the best of our knowledge, our work is the first to train a medical VLM via reinforcement learning to achieve this type of agentic and interactive visual reasoning without requiring step-by-step supervision. We have revised our claim to reflect this more precise and defensible contribution **[Line 103-107]**.

---

> ### Author Response · Authors · 2025-11-27
> **Response to Reviewer iJaK (3/3)**
>
> **Q1: What clinical workflow is referenced here?**
>
> The workflow we reference is the standard diagnostic image interpretation process employed by radiologists. A radiologist typically begins with a global overview of a scan to form a general impression. They then iteratively focus on specific regions of interest—for instance, by zooming into a suspicious nodule, adjusting window/level settings to better view tissue densities, or measuring structures. Our model's behavior—starting with a full image and then invoking the `zoom-in` tool on a specific region for closer analysis—is a simplified but direct analogy to this expert workflow.
>
> **Q2:  Implementation details of the zoom-in tool.**
>
> Thank you for these questions. We have added a comprehensive description of the tool implementation to **Appendix D.2**. Here are the specific details:
>
> 1.  **Image Resolution**: After a region is cropped, the patch is fed to the vision encoder at its **native resolution**. No additional up-sampling is performed beyond the standard pre-processing that the vision encoder requires for any input image.
>
> 2.  **Visual Token Generation**: The cropped image patch is processed by the **same pre-trained vision encoder** used for the original image.
>
> 3.  **Padding:** No special padding is applied.
>
> 4.  **Token Insertion**: The newly generated visual tokens are bracketed by special tokens`<tool_response>` and `</tool_response>`) and inserted into the model's context sequence. The model's subsequent generation is thus conditioned on its prior thoughts and this new visual information. Crucially, since these observation tokens are not generated by the policy, they are masked out during the policy loss calculation.
>
> 5.  **Latency**: We profiled the average inference latency on the OmniMedVQA dataset using H20 GPUs. The results are below. Despite the extra visual tokens, the additional latency from tool use is negligible (~1.5%). This is because autoregressive text generation is typically the main bottleneck, and in practice, the overhead from invoking the vision encoder can be further minimized with techniques like KV Caching and parallelized tool execution.
> | Model | Total Tokens (avg) | Total Latency (avg) | Extra Visual Tokens (avg) | Extra Latency (avg) |
> | --- | --- | --- | --- | --- |
> | MedVR | 169.23 | 15.21s | 84.76 | 0.23s (~1.5%) |
>
> **Q3: Regarding multiple and potentially contradictory crops.**
>
> 1.  **Possibility of Multiple Crops:** Yes, the agent can and does crop images multiple times within a successful trajectory. Our training setup allows up to 6 tool calls, indicating that multi-step visual analysis is possible. The first case study in **Figure 6** provides a clear example where the model examines both lungs sequentially.
>
> 2.  **Interpreting Non-Overlapping Crops:** Non-overlapping areas are not a contradiction but rather a sign of **systematic examination**. In the Figure 6 example, examining the right lung and then the left lung are two logical, sequential steps in a comprehensive search, mimicking how a clinician scans a chest image. A "contradiction" would be an illogical or unproductive sequence of actions. Our RL framework, which provides rewards only for successful final answers and uses CCA to reinforce consensus strategies, **inherently disincentivizes such unproductive visual search patterns** over the course of training. Trajectories with logical, sequential examinations are more likely to succeed and are therefore reinforced.
>
> **Thank you for your valuable comment again, we hope our responses can address your concern and we look forward to your further reply!**
>
> **Reference:**
>
> [1] GEMeX-RMCoT: An Enhanced Med-VQA Dataset for Region-Aware Multimodal Chain-of-Thought Reasoning. ACMMM, 2025.
>
> [2] ChestX-ray8: Hospital-Scale Chest X-Ray Database and Benchmarks on Weakly-Supervised Classification and Localization of Common Thorax Diseases. CVPR, 2017.
>
> [3] Skin lesion analysis toward melanoma detection 2018: A challenge hosted by the international skin imaging collaboration (isic). arXiv, 2019.

---

### Official Review · Reviewer_kz6o · 2025-11-10

**Soundness:** 3
**Presentation:** 3
**Contribution:** 2
**Rating:** 4
**Confidence:** 4

**Summary:**

The paper presents an RL framework that interleaves text-based reasoning with image-manipulation tools to achieve annotation-free visual reasoning in medical VLMs. Two mechanisms drive the approach: (i) Entropy-guided Visual Regrounding branches rollouts at high token-entropy steps to explore multiple regions of interest; (ii) Consensus-based Credit Assignment aggregates successful rollouts into a consensus mask and rewards trajectories whose visual footprints overlap with that mask. Experiments on three datasets, MedVR reports competitive performance.

**Strengths:**

1. The paper is well-written.
2. The motivation is sound.
3. The presented approach shows competitive performance in both in-domain and out-of-domain settings.

**Weaknesses:**

1. The paper’s comparison to prior work needs to be improved. The idea of Annotation-Free RL and Visual-Grounded Reasoning with RL has already been studied in previous works [1–7]

2. The paper lacks experiments to support the claim that the proposed method can provide robust and transparent reasoning. For example, the paper claims precise localization, but there are no quantitative grounding metrics or a human reader study to validate this. A localization-quality metric should be used. Additionally, evaluations of hallucination and robustness should be provided to support the arguments.

3. The fairness of the experiments should be clarified. It is unclear whether previous medical LLM baselines and medical RL baselines were tuned with similar rollout budgets, RL variants/settings, and training corpora. The inference settings should also be clarified to ensure a fair comparison.

[1] Right question is already half the answer: Fully unsupervised LLM reasoning incentivization. arXiv, 2025.

[2] Learning to Reason without External Rewards. arXiv, 2025.

[3] Unsupervised Post-Training for Multi-Modal LLM Reasoning via GRPO. arXiv, 2025.

[4] Grounded Reinforcement Learning for Visual Reasoning. arXiv, 2025.

[5] UniVG-R1: Reasoning Guided Universal Visual Grounding with Reinforcement Learning. arXiv, 2025.

[6] Visual Grounding for Object-Level Generalization in Reinforcement Learning. ECCV, 2024.

[7] VGR: Visual Grounded Reasoning. arXiv, 2025.

**Questions:**

Please refer to Weaknesses.

---

> ### Author Response · Authors · 2025-11-27
> **Response to Reviewer kz6o (1/2)**
>
> We sincerely thank the reviewer for the insightful feedback and constructive suggestions, which have helped us improve the quality of our manuscript. We have carefully considered all the points raised and have provided detailed responses below. We have also conducted new experiments and incorporated these results and additional discussions into the revised manuscript.
>
> **W1: Comparison to prior work**
>
> We thank the reviewer for highlighting these important related works. We agree that a more detailed comparison is beneficial and have added an expanded discussion in  **Appendix C**. Below, we summarize the key distinctions that position our work as a novel contribution.
>
> Our work differs from prior studies on Annotation-Free RL [1-3] and Visual-Grounded Reasoning [4-7] in several fundamental aspects, including problem formulation, application domain, and technical soundness.
>
> **1. Comparison with Annotation-Free RL Methods [1-3] (e.g., EMPO, INTUITOR, MM-UPT)**
>
> These methods pioneer the use of intrinsic rewards for RL without human-annotated labels. However, our work addresses a different challenge in a more complex setting.
>
> *   **Problem Definition**: The primary distinction lies in the definition of "annotation-free". Prior works [1-3] typically address tasks where the  **final answer** is unavailable, optimizing for the semantic consistency of the reasoning process itself. In contrast, our medical VQA setting provides ground-truth final answers. Our "annotation-free" challenge pertains to the **intermediate reasoning steps**. In the clinical workflow, providing a verifiable reasoning process is as crucial as the final diagnosis, yet annotating these visual-grounding steps is prohibitively expensive. MedVR is designed to learn this process without such intermediate supervision.
>
> *   **Research Domain and Modality**: The cited methods focus on text-only reasoning and are validated primarily on mathematical tasks. Our work tackles multimodal medical VQA, which introduces significant **domain and modality gaps** for general-purpose VLMs. Applying text-based intrinsic rewards directly is non-trivial and often insufficient for tasks requiring fine-grained visual evidence.
>
> *   **Framework**: Previous methods employ **textual reasoning**, often in a **single-pass** generation. MedVR utilizes an agentic reinforcement learning framework that supports **multi-turn, interleaved visual-textual reasoning**. To our knowledge, our work is the first to develop an annotation-free optimization strategy specifically for agentic RL in a multimodal context.
>
> *  **Technical Soundness**: While we also leverage concepts like entropy and voting, our implementation is tailored for interpretable, multimodal reasoning. Instead of optimizing at the token level, our EVR operates on the **semantically complete tool_call level**. Furthermore, our CCA distills a consensus mask from grounding boxes across different rollouts, providing a more interpretable supervisory signal in the visual domain.
>
> **2. Comparison with Visual-Grounded Reasoning Methods [4-7] (e.g., ViGoRL, UniVG-R1, COPL, VGR)**
>
> These works have advanced visual grounding, but they are either designed for different goals or rely on data assumptions that are impractical in the medical domain.
>
> *   **Research Task and Objective**: Methods like UniVG-R1 and COPL primarily focus on **visual grounding or localization** as the end goal. Our objective is different: we leverage the model's grounding ability as a **means to provide a verifiable visual reasoning process** to aid clinical diagnosis. The grounding is a component of a larger reasoning chain, not the final output.
>
> *   **Data Dependency**: : A critical difference is the reliance on supervision. ViGoRL and VGR depend on large-scale, human-annotated SFT datasets with fine-grained grounding information to bootstrap the model's capabilities. This is infeasible in the medical domain due to the high cost and specialized expertise required for annotation. MedVR is, to our knowledge, the first work to explore **annotation-free** visual grounded reasoning in the medical domain, directly addressing this bottleneck.
>
> *   **Methodological Focus**: The focus of ViGoRL and UniVG-R1 is largely on dataset construction and SFT, without **in-depth exploration of the RL rollout and reward design**. MedVR is the first work in visual reasoning to perform fine-grained modeling of an end-to-end agentic RL process, with novel mechanisms for uncertainty-guided exploration and consensus-based reward shaping.
>
> *  **Learning Paradigm**: The motivation of prior works is to enhance the model's grounding ability by learning from **external** high-quality SFT data. In contrast, MedVR is designed to leverage and refine the model's **internal** grounding capabilities through a self-supervisory loop within the RL process itself.
>
> We have ensured these distinctions are clearly articulated in **Appendix C** of our manuscript.

---

> ### Author Response · Authors · 2025-11-27
> **Response to Reviewer kz6o (2/2)**
>
> **W2: Localization-quality Evaluation**
>
> We thank the reviewer for this valuable suggestion. To quantitatively validate our claims of "precise localization" and "robust reasoning", we have conducted a new set of experiments to evaluate localization quality.
>
> We assessed the performance of MedVR on three distinct tasks that require precise visual grounding: General Medical VQA (on GEMEX-ThinkVG [8]), Medical Phrase Grounding (on ChestX-ray8 [9]), and Lesion Detection (on ISIC [10]). We report the mean and standard deviation of the mean Intersection over Union (mIoU) between the model's generated localization boxes and the ground-truth annotations.
>
> | Model | GEMEX-ThinkVG | ChestX-ray8 | ISIC |
> | --- | --- | --- | --- |
> | Qwen2.5-VL-7B (zero-shot) | 17.54 ± 2.13 | 36.53 ± 3.21 | 35.73 ± 1.87 |
> | **MedVR** | **59.62 ± 1.73** | **54.29 ± 1.81** | **69.12 ± 1.35** |
>
> As shown in the table, MedVR dramatically improves localization accuracy over the baseline. across all three diverse benchmarks. These results provide strong quantitative evidence for our claim of **precise localization**. The consistent, high-performance across different grounding tasks also demonstrates the **robustness**of our method. The low variance in our results further indicates **stable** performance.
>
> We have added these results to **Appendix A.2**. We believe these new experiments directly address the reviewer's concerns regarding quantitative grounding metrics.
>
> **W3: Fairness of the Experiments and Clarification of Settings**
>
> We appreciate the reviewer raising this important point. Ensuring a fair and transparent comparison is crucial.
>
> **W3.1: Fairness of the Experimental Comparison**
>
> We acknowledge that the training datasets and methods for published baselines can vary significantly, making direct comparisons challenging. For full transparency, we have added a detailed summary of the baseline models' training data and methods to **Appendix A.1**. A brief version is included here:
>
> | Model | Training Data | Training Methods | Key Benchmarks Involved in Training |
> | --- | --- | --- | --- |
> | Med-R1-2B | OmniMedVQA | RL | OmniMedVQA |
> | MedVLM-R1-2B | HuatuoGPT-Vision Eval | RL | OmniMedVQA, PMCVQA, VQA-RAD, SLAKE, PathVQA |
> | MedGemma-4B | Proprietary curated data (33M) | Pretraining, SFT, RL | PMCVQA, VQA-RAD, SLAKE and others |
> | LLava-Med-7B | PMC-15M | SFT | PMCVQA |
> | HuatuoGPT-V-7B | PubMedVsion (1.3M) | Pretraining, SFT | PMCVQA |
> | Lingshu-7B | Proprietary curated data (12M) | Pretraining, SFT, RL | PMCVQA, MedXpertQA, VQA-RAD, SLAKE, PathVQA and others |
> | MedVR (Ours) | OmniMedVQA (36k filtered) | RL | OmniMedVQA |
>
> As the table shows, many baselines were trained on significantly larger and more diverse datasets, some of which overlap with the out-of-domain test sets (e.g., Lingshu-7B).
>
> To provide a more direct and fair assessment of our method's contribution, we conducted a new **controlled experiment**. We applied our MedVR framework and a Textual RL baseline to a diverse set of backbone models (Qwen-3B/7B/32B and Lingshu-7B) using the **exact same RL settings**, including rollout budget and training corpora. This controlled setup allows us to isolate the impact of our proposed visual reasoning approach.
>
> | Model | Method | OmniMedVQA (In-domain) | PMC-VQA (OOD) | MedXpertQA (OOD) | Avg. |
> | --- | --- | --- | --- | --- | --- |
> | Qwen2.5-VL-3B | zero-shot | 63.9 | **50.6** | 21.9 | 45.5 |
> |  | Textual RL | **93.8** | 47.5 | 20.2 | 53.8 |
> |  | MedVR | 92.1 | 49.1 | **22.2** | **54.5** |
> | Qwen2.5-VL-7B | zero-shot | 64.3 | 51.2 | 22.3 | 46.0 |
> |  | Textual RL | 94.5 | 53.4 | 21.4 | 56.4 |
> |  | MedVR | **96.7** | **54.3** | **26.4** | **59.1** |
> | Qwen2.5-VL-32B | zero-shot | 69.9 | 54.1 | 24.5 | 49.5 |
> |  | Textual RL | 95.9 | 53.2 | 25.2 | 58.1 |
> |  | MedVR | **96.6** | **56.4** | **26.5** | **59.8** |
> | Lingshu-7B | zero-shot | 84.2 | 54.3 | 26.5 | 55.0 |
> |  | Textual RL | 95.7 | 53.7 | 25.1 | 58.3 |
> |  | MedVR | **96.0** | **60.8** | **26.5** | **61.1** |
>
> The results clearly demonstrate that **MedVR consistently and significantly outperforms the Textual RL baseline across all model sizes and backbones**, especially on out-of-domain (OOD) benchmarks. This provides strong evidence for the universality and effectiveness of our visual reasoning framework, validating that the performance gains are attributable to our method itself, not confounding factors. We have added these results in  **Section 4.5** of our revised manuscript.
>
> **W3.2: Illustration of the Inference Settings**
>
> We agree that these details are important for reproducibility. We have added a comprehensive description of our inference settings  for all experiments to **Appendix E**.
>
> **We hope these responses and new results have addressed the reviewer's concerns. We are confident that these additions will strengthen our paper and we look forward to your further reply!**

---

> ### Author Response · Authors · 2025-11-27
> **Reference**
>
> \[1\] Right question is already half the answer: Fully unsupervised LLM reasoning incentivization. arXiv, 2025.
>
> \[2\] Learning to Reason without External Rewards. arXiv, 2025.
>
> \[3\] Unsupervised Post-Training for Multi-Modal LLM Reasoning via GRPO. arXiv, 2025.
>
> \[4\] Grounded Reinforcement Learning for Visual Reasoning. arXiv, 2025.
>
> \[5\] UniVG-R1: Reasoning Guided Universal Visual Grounding with Reinforcement Learning. arXiv, 2025.
>
> \[6\] Visual Grounding for Object-Level Generalization in Reinforcement Learning. ECCV, 2024.
>
> \[7\] VGR: Visual Grounded Reasoning. arXiv, 2025.
>
> \[8\] GEMeX-RMCoT: An Enhanced Med-VQA Dataset for Region-Aware Multimodal Chain-of-Thought Reasoning. ACMMM, 2025.
>
> \[9\] ChestX-ray8: Hospital-Scale Chest X-Ray Database and Benchmarks on Weakly-Supervised Classification and Localization of Common Thorax Diseases. CVPR, 2017.
>
> \[10\] Skin lesion analysis toward melanoma detection 2018: A challenge hosted by the international skin imaging collaboration (isic). arXiv, 2019.

---

### Author Response · Authors · 2025-11-27
**General Response for MedVR**

We sincerely thank all the reviewers for their time and insightful feedback. We are encouraged that the reviewers recognized the **novelty, sound motivation, and clinical relevance** of our MedVR framework (kz6o, iJaK, P4Jp). The paper's **clarity and strong organization** were commended (kz6o, iJaK, P4Jp), and our core technical contributions—EVR and CCA—were highlighted as **original and effective solutions for annotation-free visual grounding** (iJaK, dA77, 9mPg). Furthermore, the framework’s **agentic design**, which mimics clinical workflows to enhance interpretability, and its consistent **state-of-the-art** performance were also positively noted (dA77, kz6o, 9mPg).

In response to the valuable concerns raised, we have conducted extensive new experiments and incorporated detailed clarifications throughout the manuscript. We summarize these key revisions below.

| Reviewer Concern | Relevant Reviewer(s) | Our Actions and Revisions |
| :--- | :--- | :--- |
| **1. Fairness and Rigor of Experimental Comparisons** | kz6o, 9mPg | We added a comprehensive summary of baseline training data for full transparency (**Appendix A.1**) and conducted **new, rigorously controlled experiments** on multiple backbones to isolate our method's contribution (**Section 4.5**). We also revised our OOD performance claims to provide a more accurate context. |
| **2. Justification and Mechanics of Proposed Methods** | dA77, 9mPg, iJaK | We clarified that Entropy-guided Visual Regrounding (EVR) is a mechanism for uncertainty-driven exploration. We provided **new empirical analysis** directly linking entropy to localization quality (**Section 4.4**) and validated Consensus-based Credit Assignment (CCA) by comparing it to a supervised upper bound (**Appendix A.3**). |
| **3. Quantitative Evaluation of Visual Grounding Quality** | kz6o, 9mPg, iJaK | We conducted **new quantitative experiments** on three diverse visual grounding benchmarks and reported mIoU results to formally validate the model's localization accuracy, demonstrating its ability to identify clinically relevant regions (**Appendix A.2**). |
| **4. Generalization and Scope of Evaluation** | dA77, 9mPg, P4Jp | We **significantly expanded our evaluation** to include free-text VQA, phrase grounding, and lesion detection tasks (**Section 4.2**). We also tested our method on general-domain benchmarks to demonstrate the broad applicability of our framework (**Appendix A.4**). |
| **5. Model Scaling and Tool Usage Analysis** | dA77, 9mPg | We performed a **new model scaling analysis** across 3B, 7B, and 32B model sizes, confirming consistent improvements (**Section 4.5**). We also provided a detailed illustration of the `zoom-in` tool's implementation and its impact (**Appendix D.2**). |
| **6. Novelty and Comparison to Prior Work** | kz6o, dA77| We added an **expanded discussion in Appendix C** with a detailed comparison to related works. We further clarified our core novelty as a practical, annotation-free framework for learning intermediate visual reasoning steps—a critical challenge in the medical domain. |
| **7. Implementation Details and Reproducibility** | kz6o, 9mPg, P4Jp, iJaK | We have provided detailed implementation specifications in a revised **Appendix D**, including our training initialization methods. We are also committed to **releasing all code, models, and scripts** upon acceptance to ensure full reproducibility. |
| **8. Clarity of Claims and Terminology** | iJaK | We have revised and clarified ambiguous terminology (e.g., "visual evidence," "verifiable") and refined our central claims regarding "grounding" and novelty to be more precise and defensible throughout the manuscript. |

We are confident that these substantial revisions, including extensive new experiments and clarifications, have thoroughly addressed the reviewers' concerns and significantly strengthened the manuscript. We thank the reviewers once again for their valuable guidance in improving our work.

Best regards,

The Authors

---

### Meta-Review · Area_Chair_F6bm · 2026-01-04

**Summary:**

The reviewers' main concerns center on a few key points: (1) Experimental Design/Sufficiency: Multiple reviewers expressed confusion about some key claims, especially about whether the localization actually works. The authors include some new results that indicate the localization is reasonable, though it would help to add stronger baselines beyond one zero-shot model. The reviewers also note that the connection between zooming-in and medical VQA isn't obvious. The authors address this with some new experiments on other medical tasks, which help address this issue. This issue would be further addressed if these types of examples are used to motivate the key ideas of the whole paper. (2) Scope of Claims: Multiple reviewers expressed confusion about whether the experiments match the authors' claims. part of this comes from the use of some vague language. The authors address some of this specific writing, though I encourage them to do a full pass on the paper to remove vague, unverified claims that may cause future readers to misunderstand the scope of this work. (3) Novelty: There are many competing works in this space right now, and at-worst the proposed method can be seen as primarily an engineering effort. To mitigate this risk, I encourage the authors to clarify their problem setup and describe how prior methods fail to meet the specific need to zoom in accurately (and adding any experiments possible to show that prior methods fail because they can't zoom in), effectively creating a stronger connection between the solution and the problem. This is a more generalizable finding than the fact that the proposed method outperforms others.

**Reviewer Concerns:**

**Addressed Concerns**:
* Authors have added experimental details (make sure future readers won't have the same confusions)
* Authors have extended the experiments according to the reviewers' suggestions

**Unaddressed Concerns**:
* Novelty of the method is hard to address, though it would help to connect the solution to challenges of the problem and highlight alternative methods' failures to address these specific challenges

**Reviewer Scores:**

R1/R2: Chance to improve
R3: No change
R4: No change

---

### Decision · Program_Chairs · 2026-01-26

Accept (Poster)